# Grouping-By-ID: Guarding Against Adversarial Domain Shifts

## Abstract

When training a deep neural network for supervised image classification, one can broadly distinguish between two types of latent features of images that will drive the classification of class $Y$. Following the notation of Gong et al. (2016), we can divide features broadly into the classes of (i) 'core' or 'conditionally invariant' features $X^{ci}$ whose distribution $P(X^{ci}|Y)$ does not change substantially across domains and (ii) 'style' or 'orthogonal' features $X^{\perp}$ whose distribution $P(X^{\perp}|Y)$ can change substantially across domains. These latter orthogonal features would generally include features such as position, rotation, image quality or brightness but also more complex ones like hair color or posture for images of persons. We try to guard against future adversarial domain shifts by ideally just using the 'conditionally invariant' features for classification. In contrast to previous work, we assume that the domain itself is not observed and hence a latent variable. We can hence not directly see the distributional change of features across different domains.

We do assume, however, that we can sometimes observe a so-called identifier or ID variable. We might know, for example, that two images show the same person, with ID referring to the identity of the person. In data augmentation, we generate several images from the same original image, with ID referring to the relevant original image. The method requires only a small fraction of images to have an ID variable.

We provide a causal framework for the problem by adding the ID variable to the model of Gong et al. (2016). However, we are interested in settings where we cannot observe the domain directly and we treat domain as a latent variable. If two or more samples share the same class and identifier, $(Y, \mathrm{ID}) = (y, \mathrm{id})$, then we treat those samples as counterfactuals under different style interventions on the orthogonal or style features. Using this grouping-by-ID approach, we regularize the network to provide near constant output across samples that share the same ID by penalizing with an appropriate graph Laplacian. This is shown to substantially improve performance in settings where domains change in terms of image quality, brightness, color changes, and more complex changes such as changes in movement and posture. We show links to questions of interpretability, fairness and transfer learning.

## 1 Introduction

Deep neural networks (DNNs) have achieved outstanding performance on prediction tasks like visual object and speech recognition (Krizhevsky et al., 2012; Szegedy et al., 2015; He et al., 2015). Issues can arise when the learned representations rely on dependencies that vanish in test distributions (e.g. see Csurka (2017) and references therein). Such domain shifts can be caused by changing conditions, e.g. color, background or location changes arising when deploying the machine learning (ML) system in production. Predictive performance is then likely to degrade. For instance, the "Russian tank legend" is an example where the training data was subject to sampling biases that were not replicated in the real world. Concretely, the story relates how a machine learning system was trained to distinguish between Russian and American tanks from photos. The accuracy was very high but only due to the fact that all images of Russian tanks were of bad quality while the photos of

American tanks were not. The system learned to discriminate between images of different qualities but would have failed badly in practice (Emspak, 2016)[1].

Hidden confounding factors like in the example above between image quality and the origin of the tank give rise to indirect associations. These are arguably one reason why deep learning requires large sample sizes as large sample sizes tend to ensure that the effect of the confounding factors averages out (although a large sample size is clearly not per se a guarantee that the confounding effect will become weaker). A large sample size is also required if one is trying to achieve invariance to known factors like translation, point of view, and rotation by using data augmentation. Another related example where human and artificial cognition deviate strongly are adversarial examples— imperceptibly but intentionally perturbed inputs that are misclassified by a ML model (Szegedy et al., 2014; Goodfellow et al., 2015). Adversarial examples do not fool humans and in general we only need to see one rotated example of the same object to achieve invariance to rotations in our perception. Our starting point is the question whether we can in a simple way mimic the human ability to learn desired invariances from a few instances of the same object and whether we can better align the features DNNs exploit with human cognition.

Considerations of fairness and discrimination might be another reason why we are interested in controlling that certain characteristics of the input data are not included in the learned representations and thus have no impact on the resulting decisions (Barocas & Selbst, 2016; Kilbertus et al., 2017). Unfortunately, existing biases in datasets used for training ML algorithms tend to be replicated in the estimated models (Bolukbasi et al., 2016). For instance, in June 2015 Google's photo app tagged two non-white people as "gorillas"—most likely because the training examples for "people" were mainly photos of white persons, making "color" predictive for the class label (Crawford, 2016; Emspak, 2016). A human would not make the same mistake after only seeing one instance of a non-white person.

Addressing the issues outlined above, we propose counterfactual regularization (CoRe) to control what latent features an estimator extracts from the input data. Conceptually, we take a causal view of the data generating process and categorize the latent data generating factors into 'conditionally invariant' (*core*) and 'orthogonal' (*style*) features, as in (Gong et al., 2016). It is desirable that a classifier uses only the *core* features as they pertain to the target of interest in a stable and coherent fashion. CoRe yields an estimator which is invariant to factors of variation corresponding to style features. Consequently, it is robust with respect to *adversarial domain shifts*, arising through arbitrarily strong interventions on the style features. CoRe relies on the fact that for certain datasets we can observe "counterfactuals" in the sense that we observe the same object under different conditions. Rather than pooling over all examples, CoRe exploits knowledge about this grouping, i.e. that a number of instances relate to the same object.

The remainder of this manuscript is structured as follows: §2 starts with two motivating examples, showing how CoRe can reduce the need for data augmentation and help predictive performance in small sample size settings. In §3 we review related work and in §4 we formally introduce counterfactual regularization, along with the CoRe estimator and theoretical insights for the logistic regression setting. In §5 we further evaluate the performance of CoRe in a variety of experiments.

## 2 TWO MOTIVATING EXAMPLES

### 2.1 GROUPING PHOTOS OF THE SAME PERSON: BETTER PREDICTIVE PERFORMANCE

The CelebA dataset (Liu et al., 2015) contains face images of celebrities. We consider the task of classifying whether a person wears glasses. Several photos of the same person are available. We use this grouping information and constrain the classification to yield the same prediction for all images belonging to the same person and sharing the same class label. We call the additional instances of the same person counterfactual (CF) observations. Figure 1a shows examples from the training set. The standard approach would be to pool all examples. The only additional information we exploit is that some observations can be grouped. We include $n = 10$ identities in the training set, resulting in a total sample size $m = 321$ as there are approximately 30 images of each person[2].

---

[1] A different version of this story can be found in Yudkowsky (2008).
[2] Additional results for $n$ ranging from 10 to 160 can be found in Figure C.7b.

(a) Grouping-by-ID with ID=identity.  (b) Grouping-by-ID with ID=original image.

Figure 1: Examples from a) the subsampled CelebA dataset and b) the augmented MNIST dataset. Connected images are counterfactual examples as they share the same realization of the ID which is the identity of the person in a) and the original image used for data augmentation in b). The comparison is a training of exactly the same network architecture that does not make use of the grouping information but using a standard ridge penalty. In a) exploiting the grouping information reduces the test error by 32% compared to pooling over all samples. In b) the test error on rotated digits is reduced by 50%.

Exploiting the group structure reduces the average test error from 24.76% to 16.89%, i.e. by approx. 32%, compared to the estimator which just pools all images and uses a standard ridge penalty for the cofficients[3].

## 2.2 GROUPING AUGMENTED IMAGES BY ORIGINAL: MORE SAMPLE EFFICIENT

A different use case of CORE is to make data augmentation more efficient in terms of the required samples. In data augmentation, one creates additional samples by modifying the original inputs, e.g. by rotating, translating, or flipping the images (Schölkopf et al., 1996). In other words, additional samples are generated by interventions on style features. Using this augmented data set for training results in invariance of the estimator with respect to the transformations (style features) of interest. For CORE we can use the grouping information that the original and the augmented samples belong to the same object. This enforces the invariance with respect to the style features more strongly compared to normal data augmentation which just pools all samples. We assess this for the style feature "rotation" on MNIST (LeCun & Cortes, 2010) and only include $c = 100$ augmented training examples for $n = 10000$ original samples, resulting in a total sample size of $m = 10100$. The degree of the rotations is sampled uniformly at random from $[35, 70]$. Figure 1b shows examples from the training set. By using CORE the average test error on rotated examples is reduced from 32.86% to 16.33%, around half its original value[4].

## 3 RELATED WORK

Perhaps most similar to this work in terms of their goals are the work of Gong et al. (2016) and Domain-Adversarial Neural Networks (DANN) proposed in Ganin et al. (2016), an approach motivated by the work of Ben-David et al. (2007). While our approach requires grouped observations, both of these works rely on unlabeled data from the target task being available.

The main idea of Ganin et al. (2016) is to learn a representation that contains no discriminative information about the origin of the input (source or target domain). This is achieved by an adversarial training procedure: the loss on domain classification is maximized while the loss of the target prediction task is minimized simultaneously. In contrast, we do not assume that we have data from different domains but just different realizations of the same object under different interventions.

The data generating process assumed in Gong et al. (2016) is similar to our model, introduced in §4.2 where we detail the similarities and differences between the models (cf. Figure 2). Gong et al. (2016) identify the conditionally independent features by adjusting a transformation of the

---

[3]Details on the architecture can be found in Table C.1. Using ImageNet pre-trained features from Inception V3 does not yield lower error rates.

[4]Additional results for $n \in \{1000, 10000\}$ and $c$ ranging from 100 to 5000 can be found in Figure C.8.

variables to minimize the squared MMD distance between distributions in different domains[5]. The fundamental difference to our approach is that we use a different data basis. The domain identifier is explicitly observable in Gong et al. (2016), while it is latent in our approach. In contrast, we exploit presence of an identifier variable ID to penalize the classifier using any latent features outside the set of conditionally independent features.

Causal modeling has related aims to the setting of transfer learning and guarding against adversarial domain shifts. Specifically, causal models have the defining advantage that the predictions will be valid even under arbitrarily large interventions on all predictor variables (Haavelmo, 1944; Aldrich, 1989; Pearl, 2009; Schölkopf et al., 2012; Peters et al., 2016; Zhang et al., 2013; 2015; X. Yu, 2017; M. Rojas-Carulla, 2017; Magliacane et al., 2017). There are two difficulties in transferring these results to the setting of adversarial domain changes in image classification. The first hurdle is that the classification task is typically anti-causal since the image we use as a predictor is a descendant of the true class of the object we are interested in rather than the other way around. The second challenge is that we do not want to guard against arbitrary interventions on any or all variables but only would like to guard against a shift of the style features. It is hence not immediately obvious how standard causal inference can be used to guard against large domain shifts.

Recently, various approaches have been proposed that leverage causal motivations for deep learning or use deep learning for causal inference. In all of the following methods, the goals and the settings are different from ours. Specifically, the setting of anti-causal prediction and non-ancestral interventions on style variables is not considered. Various approaches focus on cause-effect inference where the goal is to find the causal relation between two random variables, $X$ and $Y$ (Lopez-Paz et al., 2017; Lopez-Paz & Oquab, 2017; Goudet et al., 2017). Lopez-Paz et al. (2017) propose the Neural Causation Coefficient (NCC) to estimate the probability of $X$ causing $Y$ and apply it to finding the causal relations between image features. Specifically, the NCC is used to distinguish between features of objects and features of the objects' contexts. Lopez-Paz & Oquab (2017) note the similarity between structural equation modeling and CGANs (Mirza & Osindero, 2014). One CGAN is fitted in the direction $X \rightarrow Y$ and another one is fitted for $Y \rightarrow X$. Based on a two-sample test statistic, the estimated causal direction is returned. Goudet et al. (2017) use generative neural networks for cause-effect inference, to identify $v$-structures and to orient the edges of a given graph skeleton. Bahadori et al. (2017) devise a regularizer that combines an $\ell_1$ penalty with weights corresponding to the estimated probability of the respective feature being causal for the target. The latter estimates are obtained by causality detection networks or scores such as estimated by the NCC. Besserve et al. (2017) draw connections between GANs and causal generative models, using a group theoretic framework. Kocaoglu et al. (2017) propose causal implicit generative models to sample from conditional as well as interventional distributions, using a conditional GAN architecture (CausalGAN). The generator structure needs to inherit its neural connections from the causal graph, i.e. the causal graph structure must be known. Louizos et al. (2017) propose the use of deep latent variable models and proxy variables to estimate individual treatment effects.

Kilbertus et al. (2017) exploit causal reasoning to characterize fairness considerations in machine learning. Distinguishing between the protected attribute and its proxies, they derive causal non-discrimination criteria. The resulting algorithms avoiding proxy discrimination require classifiers to be constant as a function of the proxy variables in the causal graph, thereby bearing some structural similarity to our style features.

Distinguishing between core and style features can be seen as some form of disentangling factors of variation. Estimating disentangled factors of variation has gathered a lot of interested in the context of generative modeling (Higgins et al., 2017; Chen et al., 2016; Bouchacourt et al., 2017). For example, Matsuo et al. (2017) propose a "Transform Invariant Autoencoder" where the goal is to reduce the dependence of the latent representation on a specified transform of the object in the original image. Specifically, Matsuo et al. (2017) predefine location as the orthogonal style feature $X^{\perp}$ and the goal is to learn a latent representation that does not include $X^{\perp}$. Here, we do not predefine which features are in $X^{\perp}$. It could be location but also image quality, posture, brightness, background and contextual information. Additionally, the approach in Matsuo et al. (2017) cannot effectively deal

---

[5]The distinction between 'conditionally independent' features and 'conditionally transferable' (which is the former modulo location and scale transformations) is for our purposes not relevant as we never make a linearity assumption and the core or conditionally independent features would then always refer to the identity of the object and not its location, scale or any other style feature.

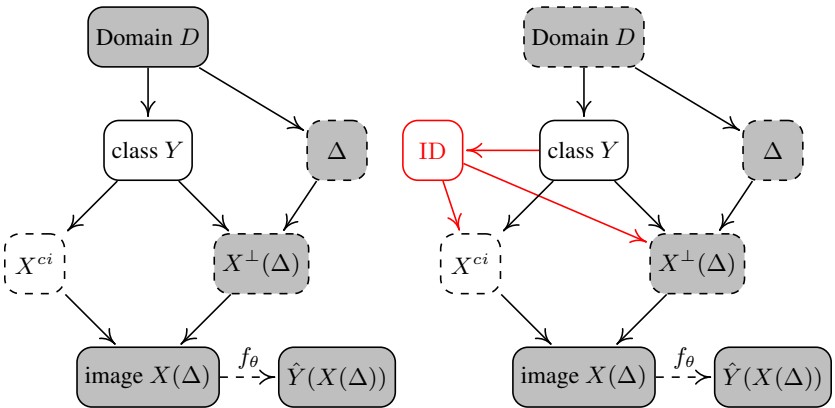

Figure 2: Left: data generating process for the considered model as in Gong et al. (2016), where the effect of the domain on the orthogonal features $X^\perp$ is mediated via unobserved noise $\Delta$. Right: our setting. The domain itself is unobserved but we can now observe the ID variable we use for grouping.

with a confounding situation where the distribution of the style features differs conditional on the class (this is a natural restriction as the class label is not even observed in the autoencoder setting). As in CORE, Bouchacourt et al. (2017) exploit grouped observations. In a variational autoencoder framework, they aim to separate style and content—they assume that samples within a group share a common but unknown value for one of the factors of variation while the style can differ. Here we try to solve a classification task directly without estimating the latent factors explicitly as in a generative framework.

## 4 COUNTERFACTUAL REGULARIZATION

We first describe the standard notation for classification before developing a causal graph that allows us to compare the setting of adversarial domain shifts to transfer learning, domain adaptation and adversarial examples.

### 4.1 NOTATION FOR STANDARD CLASSIFICATION

Let $Y \in \mathcal{Y}$ be a target of interest. Typically $\mathcal{Y} = \mathbb{R}$ for regression or $\mathcal{Y} = \{1, \ldots, K\}$ in classification with $K$ classes. Let $X \in \mathbb{R}^p$ be a predictor, for example the $p$ pixels of an image. The prediction $\hat{y}$ for $y$, given $X = x$, is of the form $f_\theta(x)$ for a suitable function $f_\theta$ with parameters $\theta \in \mathbb{R}^d$, where the parameters $\theta$ correspond to the weights in a DNN. For regression, $f_\theta(x) \in \mathbb{R}$, whereas for classification $f_\theta(x)$ corresponds to the conditional probability distribution of $Y \in \{1, \ldots, K\}$. Let $\ell$ be a suitable loss that maps $y$ and $\hat{y} = f_\theta(x)$ to $\mathbb{R}^+$. A standard goal is to minimize the expected loss or risk

$$L(\theta) = E\Big[\ell(Y, f_\theta(X))\Big].$$

Let $(x_i, y_i)$ for $i = 1, \ldots, n$ be the samples that constitute the training data and $\hat{y}_i = f_\theta(x_i)$ the prediction for $y_i$. A standard approach to parameter estimation is penalized empirical risk minimization, where we choose the weights or parameters as $\hat{\theta} = \text{argmin}_\theta \, L_n(\theta)$, with the empirical loss given by $L_n(\theta) = \frac{1}{n} \sum_{i=1}^n \ell(y_i, f_\theta(x_i)) + \lambda \cdot \text{pen}(\theta)$, where the penalty $\text{pen}(\theta)$ could be a ridge penalty or penalties that exploit underlying geometries such as the Laplacian regularized least squares (Belkin et al., 2006).

### 4.2 CAUSAL GRAPH

The full structural model for all variables is shown in the right panel of Figure 2. The domain variable $D$ is latent, in contrast to Gong et al. (2016). We add the ID variable (identity of a person, for example), whose distribution can change conditional on class $Y$. The ID variable is used to

group observations, see Section 4.4, and can be assumed to be latent in the setting of Gong et al. (2016).

The rest of the graph is in analogy to Gong et al. (2016). The prediction is anti-causal, that is the predictors $X$ that we use for $\hat{Y}$ are non-ancestral to $Y$. In other words, the class label is causal for the image and not the other way around. The causal effect from the class label $Y$ on the image $X$ is mediated via two types of latent variables: the so-called *core* or 'conditionally invariant' features $X^{ci}$ and the orthogonal or *style* features $X^{\perp}$. The distinguishing factor between the two is that external interventions $\Delta$ are possible on the *style* features but not on the *core* features. If the interventions $\Delta$ have different distributions in different domains, then the distribution $P(X^{ci}|Y)$ is constant across domains while $P(X^{\perp}|Y)$ can change across domains. The style features $X^{\perp}$ and $Y$ are confounded, in other words, by the latent domain $D$. In contrast, the *core* or 'conditionally invariant' features satisfy $X^{ci} \perp\!\!\!\perp D|Y$. The dimension of $X^{ci}$ is chosen maximally large such that this conditional independence is still true. The *style* variable can include point of view, image quality, resolution, rotations, color changes, body posture, movement etc. and will in general be context-dependent[6]. The style intervention variable $\Delta$ influences both the latent style $X^{\perp}$, and hence also the image $X$. In potential outcome notation, we let $X^{\perp}(\Delta = \delta)$ be the style under intervention $\Delta = \delta$, $X(Y, \text{ID}, \Delta = \delta)$ the image for class $Y$, identity ID and style intervention $\Delta$ and this sometimes abbreviated as $X(\Delta = \delta)$ for notational simplicity. Finally, $f_{\theta}(X(\Delta = \delta))$ is the prediction under the style intervention $\Delta = \delta$. For a formal justification of using a causal graph and potential outcome notation simultaneously see Richardson & Robins (2013).

### 4.3 DOMAIN ADAPTATION, ADVERSARIAL EXAMPLES AND ADVERSARIAL DOMAIN SHIFTS

In this work, we are interested in guarding against adversarial domain shifts. We use the causal graph to explain the related but not identical goals of domain adaptation, transfer learning and guarding against adversarial examples.

(i) **Domain adaptation and transfer learning.** Assume we have $J$ different domains, each with a new distribution $F_j$ for the interventions $\Delta$ (or more generally of the joint distribution of $(Y, \Delta)$). The shift of $F_j$ for different domains $j = 1, \ldots, J$ causes a shift in both the distribution of $X$ and in the conditional distribution $Y|X$. If we consider domain adaptation and transfer learning together, their goal is generally to give the best possible prediction $\hat{Y}_j(x)$ in each domain $j = 1, \ldots, J$.

(ii) **Standard adversarial examples.** The setting of adversarial examples in the sense of Szegedy et al. (2014) and Goodfellow et al. (2015) can also be described by the causal graph above by using $X^{\perp}(\Delta) = \Delta$ and identifying $X^{\perp}$ with pixel-by-pixel additive effects. The magnitude of the intervention $\Delta$ is then typically assumed to be within an $\epsilon$-ball in $\ell_q$-norm around the origin, with $q = \infty$ or $q = 2$ for example. If the input dimension is large many imperceptible changes in the coordinates of $X$ can cause a large change in the output, leading to a misclassification of the sample. The goal is to devise a classification in this graph that minimizes the adversarial loss

$$E\Big[ \max_{\Delta \in \mathbb{R}^q: \, \|\Delta\|_q \leq \epsilon} \ell\Big(Y, f_{\theta}\big(X(\Delta)\big)\Big)\Big], \tag{1}$$

where $X(\Delta)$ is the image under the intervention $\Delta$ and $\hat{Y} = f_{\theta}(X(\Delta))$ is the estimated conditional distribution of $Y$, given the image under the chosen interventions.

(iii) **Adversarial domain shifts.** Here we are interested in arbitrarily strong interventions $\Delta \in \mathbb{R}^q$ on the style features $X^{\perp}$, which are not known explicitly in general. Analogously to (1), the adversarial loss under arbitrarily large style interventions is

$$L_{adv}(\theta) = E\Big[ \max_{\Delta \in \mathbb{R}^q} \ell\Big(Y, f_{\theta}\big(X(\Delta)\big)\Big)\Big]. \tag{2}$$

In contrast to (1) the interventions can be arbitrarily strong but we assume that the style features $X^{\perp}$ can only change certain aspects of the image, while other aspects of the image (mediated by the core features) cannot be changed. In contrast to Ganin et al. (2016), we use the term

---

[6]The type of features we regard as style and which ones we regard as core features can conceivably change depending on the circumstances—for instance, is the color "gray" an integral part of the object "elephant" or can it be changed so that a colored elephant is still considered to be an elephant?

"adversarial" to refer to adversarial interventions on the style features, while the notion of "adversarial" in domain adversarial neural networks describes the training procedure. Nevertheless, the motivation of Ganin et al. (2016) is equivalent to ours—that is, to protect against shifts in the distribution(s) of test data which we characterize by distinguishing between core and style features.

## 4.4 COUNTERFACTUAL OBSERVATIONS / GROUPING

The classical problem of causal inference is that we can never observe a counterfactual. For instance, we can only see the health outcome $Z$ if we take a medicine, $T = 1$, or not, $T = 0$, but we can never see both health outcomes simultaneously. The counterfactual in this context would be an observation where we change the treatment but hold all observed and unobserved confounders constant. If the treatment $T$ changes while all other variables are kept constant, we could just read off the treatment effect as $Z(T = 1) - Z(T = 0)$ if $Z$ is the health outcome of interest. Observing such counterfactuals is in general impossible as we can either observe the outcome under treatment or under no treatment but not both.

Here, we use the term counterfactual for a situation where we keep class label $Y$ and ID constant but allow the value of the style intervention $\Delta$ to change. The new value of $\Delta$ could be a do-intervention (as when explicitly rotating an image in data augmentation) or it could be a noise-intervention by sampling a new realization of $\Delta$. The style intervention $\Delta$ takes the same role as the treatment $T$ in the previous medical example. In contrast to the medical example, however, counterfactuals are conceivable for image analysis as we can see the same object $(Y, \text{ID})$ under different conditions ('treatments') $\Delta$.

As an example, if $Y$ is the binary variable whether a person wears glasses and ID is the identity of a person, then $\Delta$ corresponds to all other variables that determine the different images of the same person (either consistently wearing glasses or not) and includes background, posture, viewing angle, image quality, etc.

In further contrast to the medical setting, we are not interested primarily in the 'treatment effect' of the style intervention $\Delta$ but we merely use it to implicitly rule out parts of the feature space for classification. We know that any 'treatment effect' of $\Delta$ occurs in the space of the style or orthogonal features $X^{\perp}$ and not in the 'conditionally invariant' space $X^{ci}$ and we would thus like to penalize any change in the classification under different style interventions $\Delta$ but constant class and identity $(Y, \text{ID})$.

Notationally, we have for sample $i \in \{1, \ldots, n\}$ with class label and identifier $(Y, \text{ID}) = (y_i, \text{id}_i)$, $m_i$ different images $x(y_i, \text{id}_i, \Delta_{i,j})$ for $j = 1, \ldots, m_i$ under different (unobserved) values of $\Delta_{i,1}, \ldots, \Delta_{i,m_i}$. Let $m = \sum_i m_i$ denote the total number of samples and $c = m - n$, the number of counterfactual observations. Denote the $j$-th observation of sample $i$, by $x_{i,j} \in \mathbb{R}^p$. Typically $m_i = 1$ for most samples and occasionally $m_i \geq 2$.

### 4.4.1 STANDARD APPROACH: POOLED ESTIMATOR

The standard approach is to simply pool over all available observations, ignoring any grouping information that might be available. The pooled estimator thus treats all examples identically by summing over the loss as

$$\hat{\theta}^{pool} = \text{argmin}_{\theta} \frac{1}{m} \sum_{i=1}^{n} \sum_{j=1}^{m_i} \left[ \ell\big(y_i, f_{\theta}(x_{i,j})\big) \right] + \lambda \cdot \text{pen}(\theta),$$

where $\text{pen}(\theta)$ could be a ridge penalty. The pooled estimator in all examples is always the ridge estimator with a cross-validated choice of the penalty parameter. The adversarial loss of the pooled estimator will in general be infinite; see §4.6 for a concrete example. Using Figure 2, one can show that the pooled estimator will work well in terms of the adversarial loss $L_{adv}$ if both (i) $Y \perp\!\!\!\perp X | X^{ci}$ and (ii) $Y \not\perp\!\!\!\perp X^{ci} | X^{\perp}$. The first condition (i) implies that if the estimator learns to extract $X^{ci}$ from the image $X$, there is no further information in $X$ that explains $Y$ and, therefore, the direction corresponding to $X^{\perp}$ is not required for predicting $Y$. The second condition (ii) is fulfilled if the relations between $Y$, $X^{ci}$, and $X^{\perp}$ are not deterministic. Intuitively, it ensures that $X^{\perp}$ cannot replace $X^{ci}$ in the first condition. From (i) and (ii), we see that the pooled estimator will work well

in terms of the adversarial loss $L_{adv}$ if (a) the edge from $X^\perp$ to $X$ is absent or if (b) both the edge from $D$ to $X^\perp$ and the edge from $Y$ to $X^\perp$ are absent (cf. Figure 2).

## 4.5 CORE ESTIMATOR

In order to minimize the adversarial loss (2) we have to ensure $f_\theta(x(\Delta))$ is as constant as possible as a function of $\Delta$ for all $x \in \mathbb{R}^p$. Let $I$ be the *invariant parameter space*

$$I := \{\theta : f_\theta(x(\Delta)) \text{ is constant as function of } \Delta \text{ for all } x \in \mathbb{R}^p\}$$
$$= \{\theta : f_\theta(x) = f_\theta(x^{ci}) \text{ is a function of the core features } x^{ci} \in \mathbb{R}^p \text{ only.}\}.$$

For all $\theta \in I$, the adversarial loss (2) is identical to the loss under no interventions at all. More precisely, let $X$ be a shorthand notation for $X(\Delta = 0)$, the images in absence of external interventions:

$$\text{if } \theta \in I, \text{ then } \quad E\Big[\max_{\Delta \in \mathbb{R}^q} \ell\Big(Y, f_\theta(X(\Delta))\Big)\Big] \; = \; E\Big[\ell\Big(Y, f_\theta(X)\Big)\Big].$$

The optimal predictor in the invariant space $I$ is

$$\theta^* \; = \; \text{argmin}_\theta \; E\Big[\ell(Y, f_\theta(X))\Big] \text{ such that } \theta \in I. \tag{3}$$

If $f_\theta$ is only a function of the core features $X^{ci}$, then $\theta \in I$. The challenge is that the core features are not directly observable and we have to infer the invariant space $I$ from data. To get an approximation to the optimal invariant parameter vector (3), we use empirical risk minimization:

$$\hat{\theta}^{core} \; = \; \text{argmin}_\theta \; \frac{1}{n} \sum_{i=1}^{n} \sum_{j=1}^{m_i} \ell\Big(y_i, f_\theta(x_{i,j})\Big) \text{ such that } \theta \in I_n, \tag{4}$$

where the first part is the empirical version of the expectation in (3). The unknown invariant parameters space $I$ is approximated by an empirically invariant space $I_n$, defined as

$$I_n := \{\theta : \sum_{i=1}^{n} \sigma_i^2(\theta) \leq \tau\},$$

where $\sigma_i^2(\theta)$ is the variance of $f_\theta(x_{i,j})$ when varying $j = 1, \ldots, m_i$ for a fixed value of $i$ and $\tau \geq 0$ is a regularization constant. Setting $\tau = 0$ is equivalent to demanding that the estimated predictions for the class labels are identical across all $m_i$ counterfactuals of image $i$, while slightly larger values of $\tau$ allow for some small degree of variations. For all values $\tau \geq 0$ the true invariant space $I$ is a subset of the empirically invariant subspace $I_n$, that is $I \subseteq I_n$. Under the right assumptions we get $I_n = I$ for $n \to \infty$. We return to this question in §4.6. One can equally use the Lagrangian form of the constrained optimization in (4), with a penalty parameter $\lambda$ instead of a constraint $\tau$ to get

$$\hat{\theta}^{core} \; = \; \text{argmin}_\theta \; \frac{1}{n} \sum_{i=1}^{n} \sum_{j=1}^{m_i} \ell\Big(y_i, f_\theta(x_{i,j})\Big) + \lambda \cdot \text{pen}_{\text{ID}}(\theta), \tag{5}$$

where $\text{pen}_{\text{ID}}(\theta) = \tilde{f}_\theta^t L_{\text{ID}} \tilde{f}_\theta$, and $\tilde{f}_\theta \in \mathbb{R}^m$ is the value of $f_\theta(x_{i,j})$ at all $m = \sum_{i=1}^{n} m_i$ observations. The matrix $L_{\text{ID}}$ is a graph Laplacian (Belkin et al., 2006), where the underlying graph has $n$ connectivity components as all samples that have the same ID are connected by an edge and form fully connected connectivity components. The graph Laplacian regularization is identical to penalizing the sum over the variances $\sigma_i^2(\theta)$. The graph for the underlying regularization is formed in the sample space and induced by the identifier variable ID, in contrast to graphs formed in feature space as in Sandler et al. (2009), where prior knowledge is used to form the graph by connecting features that share similar characteristics.

We show in §C.1 that the outcome does not depend strongly on the chosen value of the penalty $\lambda$ and the experiments show that it is crucial to define the graph in terms of the identifier variable ID. Other regularizations do not perform nearly as well when trying to guard against adversarial domain shifts.

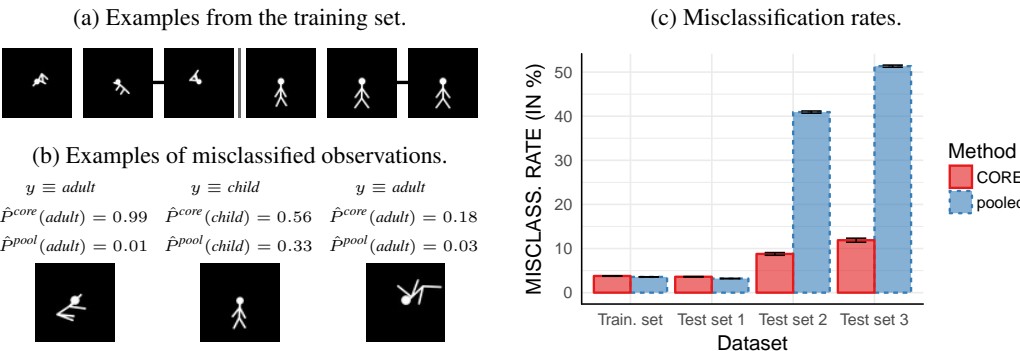

Figure 3: a) Examples from the stickmen training set. The first three images from the left have $y \equiv child$; the remaining three images have $y \equiv adult$. Connected images are counterfactual examples. b) Misclassified observations from test set 2. c) Misclassification rates for $c = 50$. Results for $c \in \{20, 500, 2000\}$ can be found in Figure C.10.

## 4.6 THEORETICAL RESULTS

In §A we analyze the adversarial loss, defined in Eq. (2), for the pooled and the CORE estimator in a one-layer network for binary classification (logistic regression). Here, we briefly sketch the result while all details are given in §A. Assume the structural equation for the image $X \in \mathbb{R}^p$ is linear in the style features $X^\perp \in \mathbb{R}^q$ (with generally $p \gg q$), the interventions are additive and we use logistic regression to predict a class label $Y \in \{-1, 1\}$. Under suitable assumptions (cf. Assumption 1), the pooled estimator has infinite adversarial loss while the adversarial loss of the CORE estimator converges to the optimal adversarial loss as $n \to \infty$.

## 5 EXPERIMENTS

We perform an array of different experiments: in §5.1 and §5.2 we study how CORE can handle confounded training data sets and changing style features in test distributions. For the assessment we explicitly control the level of confounding. In §5.3, we consider classifying elephants and horses where $X^\perp \equiv color$. In §B, we include two additional experiments: in the first one, $Y \equiv gender$ and $X^\perp \equiv wearing\ glasses$; in the second one, $Y \equiv wearing\ glasses$ and $X^\perp \equiv brightness$. Additional experimental results for the settings introduced in §2 can be found in §C.2 and §C.3. A TensorFlow (Abadi et al., 2015) implementation of CORE will be made available as well as further code necessary to reproduce the experiments. In addition to the details provided below, information on the employed architectures can be found in §C.7. An open question is how to set the value of the tuning parameter $\tau$ or the penalty $\lambda$ in Lagrangian form. We show in §C.1 that performance is typically not very sensitive to the choice of $\lambda$.

## 5.1 STICKMEN IMAGE-BASED AGE CLASSIFICATION

In this example we consider synthetically generated stickmen images (cf. Figure 3a). The target of interest is $Y \in \{adult, child\}$ and $X^{ci} \equiv height$. The class $Y$ is causal for height and height cannot be easily intervened on, so we consider it to be a core feature—it is a robust predictor for differentiating between children and adults. Additionally, there is a dependence between age and $X^\perp \equiv movement$ in the training dataset which arises through the hidden common cause $D \equiv place\ of\ observation$. The data generating process is illustrated in Figure C.9. For instance, the images of children might mostly show children playing while the images of adults typically show them in more "static" postures. If the learned model exploits this dependence for predicting $Y$, it will fail when presented images of, say, dancing adults.

Figure 3a shows examples from the training set where large movements are associated with children and small movements are associated with adults. Test set 1 follows the same distribution. In test sets 2 and 3 $X^\perp$ is intervened on such that the edge from $D$ to $X^\perp$ is removed and the dependence between $Y$ and $X^\perp$ vanishes. In test sets 2 and 3 large movements are associated with both children and adults, while the movements are heavier in test set 3 than in test set 2. Figure C.10 shows examples from all test sets. Figure 3c shows misclassification rates for CORE and the pooled estimator

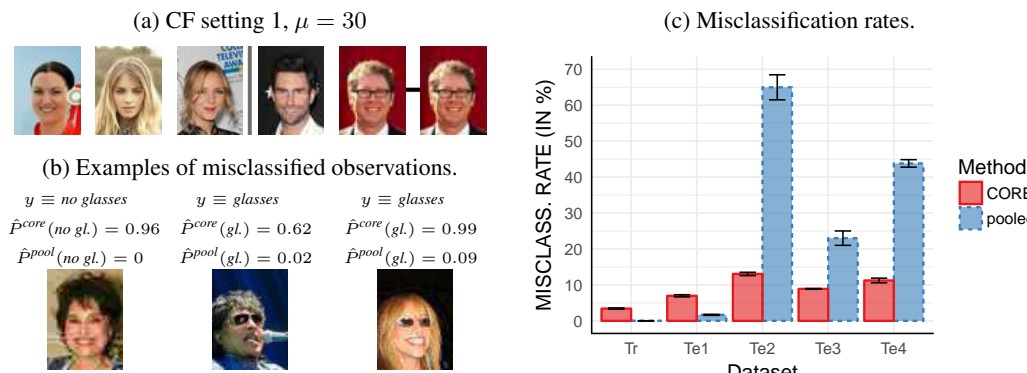

Figure 4: a) Examples from the CelebA image quality dataset. The first three images from the left have $y \equiv no\ glasses$; the remaining three images have $y \equiv glasses$. Connected images are counterfactual examples. b) Misclassified examples from the test sets. c) Misclassification rates for $\mu = 30$ and $c = 5000$. Results for different counterfactual settings and $\mu \in \{30, 40, 50\}$ can be found in Figure C.12.

for $c = 50$ with a total sample size of $m = 20000$. For as few as 50 counterfactual observations, CORE succeeds in achieving good predictive performance on test sets 2 and 3 where the pooled estimator fails (test errors $> 40\%$). These results suggest that the learned representation of the pooled estimator uses movement as a predictor for age while CORE does not use this feature due to the counterfactual regularization. Importantly, including more counterfactual examples would not improve the performance of the pooled estimator as these would be subject to the same bias and hence also predominantly have examples of heavily moving children and "static" adults (also see Figure C.10 which shows results for $c \in \{20, 500, 2000\}$).

## 5.2 EYEGLASSES DETECTION: IMAGE QUALITY INTERVENTION

As in §2.1, we use the CelebA dataset and consider the problem of classifying whether the person in the image is wearing eyeglasses. Here, $X^{\perp}$ is the quality of the image which differs conditional on $Y$[7]—if the image shows a person wearing glasses, the image quality tends to be lower. This setting mimics the confounding that occurred in the Russian tank legend (cf. §1). The strength of the image quality intervention is governed by sampling the new image quality as a percentage of the original image's quality from a Gaussian distribution $\mathcal{N}(\mu = 30, \sigma = 10)$. Images of people without glasses are not changed. Thus, we only have counterfactual observations for $Y \equiv glasses$. Figure 4a shows examples from the training set. Here, we use as the counterfactual observation the same image but with a newly sampled image quality value from $\mathcal{N}(30, 10)$. We call using the same image as a counterfactual "CF setting 1". Two alternatives for constructing counterfactual observations for this setting are discussed in §B.2.1. Here, $c = 5000$ and $m = 20000$.

Figure 4c shows misclassification rates for CORE and the pooled estimator on different test sets. Examples from all test sets can be found in Figure C.11. Test set 1 follows the same distribution as the training set. In test set 2 the class of the quality intervention is reversed, i.e. the quality of images showing people without glasses tends to be lower. In test set 3 all images are left unchanged and in test set 4 the quality of all images is decreased. First, we notice that the pooled estimator performs better than CORE on test set 1. This can be explained by the fact that it can exploit the predictive information contained in an image's quality while CORE is restricted not to do so. Second, we observe that the pooled estimator does not perform well on test sets 2–4 as its learned representation seems to use the image's quality as a predictor for the target. In contrast, the predictive performance of CORE is hardly affected by the changing image quality distributions. More experimental details are provided in §C.5. Results for quality interventions of different strengths ($\mu \in \{30, 40, 50\}$) are shown in Figure C.12.

---

[7]In §B.2 we additionally look at the case where the brightness distribution differs conditional on $Y$.

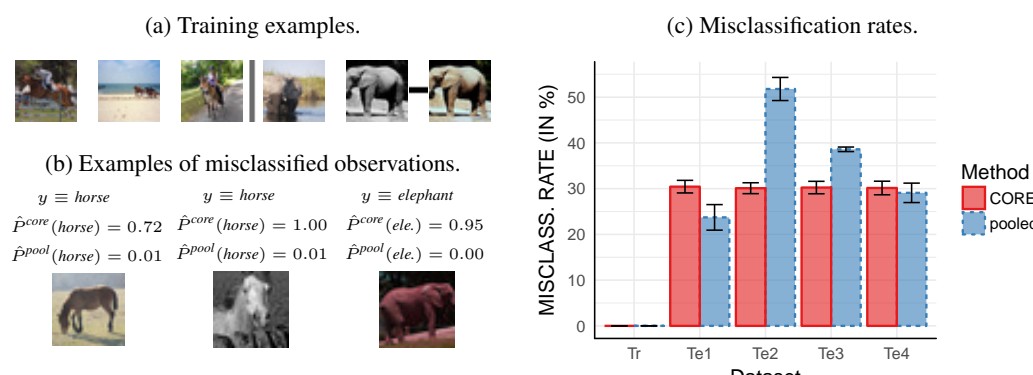

Figure 5: a) Examples from the subsampled and augmented AwA2 dataset. The first three images from the left shows horses, the remaining three images show elephants. Connected images are counterfactual examples. b) Misclassified examples from the test sets. c) Misclassification rates.

### 5.3 ELMER THE ELEPHANT

In this example, we want to assess whether invariance with respect to $X^{\perp} \equiv color$ can be achieved. In the children's book "Elmer the elephant"[8] one instance of a colored elephant suffices to recognize it as being an elephant, making the color "gray" no longer an integral part of the object "elephant". Motivated by this process of concept formation, we would like to assess whether CORE can exclude "color" from its learned representation by including a few counterfactuals of different color.

We work with the "Animals with attributes 2" (AwA2) dataset (Xian et al., 2017) and consider classifying images of horses and elephants. The data generating process is illustrated in Figure C.14. We include counterfactual examples by adding grayscale images for $c = 250$ images of elephants, i.e. counterfactuals are only available for one class and the shift in color is quite subtle. The total sample size is 1850.

Figure 5a shows examples from the training set and Figure 5c shows misclassification rates for CORE and the pooled estimator on different test sets. Examples from all test sets can be found in Figure C.13. Test set 1 contains original, colored images only. In test set 2 images of horses are in grayscale and the colorspace of elephant images is modified, effectively changing the color gray to red-brown. Test set 3 contains grayscale images only and in test set 4 the colorspace of all images is shifted towards red. The details are given in §C.6 . We observe that the pooled estimator does not perform well on test sets 2 and 3 as its learned representation seems to exploit the fact that "gray" is predictive for the target in the training set. Using this information helps its predictive accuracy on test set 1. In contrast, the predictive performance of CORE is hardly affected by the changing color distributions.

It is noteworthy that a colored elephant can be recognized as an elephant by adding a few examples of a grayscale elephant to the very lightly colored pictures of natural elephants. If we just pool over these examples, there is still a strong bias that elephants are gray. The CORE estimator, in contrast, demands invariance of the prediction for instances of the same elephant and we can learn color invariance with a few added grayscale images.

While a thorough analysis in terms of fairness considerations is beyond the scope of this work, we would like to draw the following connection. If "color" was a protected attribute or a proxy for one, CORE would satisfy fairness in the sense that it would not include it in its learned representation. In contrast, there is no way to avoid that the pooled estimator extracts and uses "color" for its decisions.

## 6 CONCLUSION

Distinguishing the latent features in an image into *core* and *style* features, we have proposed counterfactual regularization (CORE) to achieve robustness with respect to arbitrarily large interventions on the style or conditionally invariant features. The main idea of the CORE estimator is to exploit the fact that we often have instances of the same object in the training data. By demanding invariance of

---

[8]https://en.wikipedia.org/wiki/Elmer_the_Patchwork_Elephant

the classifier amongst a group of instances that relate to the same object, we can achieve invariance of the classification performance with respect to adversarial interventions on style features such as image quality, fashion type, color, or body posture. The training also works despite sampling biases in the data.

There are two main applications areas. If the style features are known explicitly, we can achieve the same classification performance as standard data augmentation approaches but using fewer instances which, on top, do not have to be carefully balanced in the training data. Perhaps more interestingly, if the style features are unknown, the regularization of CORE avoids usage of them automatically by penalizing features that vary strongly between different instances of the same object in the training data.

An interesting line of work would be to use larger models such as Inception or large ResNet architectures (Szegedy et al., 2015; He et al., 2016). These models have been trained to be invariant to an array of explicitly defined style features. In §B.1 we include results which show that using Inception V3 features does not guard against interventions on more implicit style features. We would thus like to assess what benefits CORE can bring for training Inception-style models end-to-end, both in terms of sample efficiency and in terms of generalization performance.

While we showed some examples where the necessary grouping information is available, an interesting possible future direction would be to use video data since objects display temporal constancy and the temporal information can hence be used for grouping and counterfactual regularization. Potentially an analogous approach could also help to debias word embeddings.

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

# SUPPLEMENTARY MATERIAL

## A    LOGISTIC REGRESSION

Assume the structural equation for the image $X \in \mathbb{R}^p$ is linear in the style features $X^\perp \in \mathbb{R}^q$ (with generally $p \gg q$) and we use logistic regression to predict a class label $Y \in \{-1, 1\}$. Let the interventions $\Delta \in \mathbb{R}^q$ act additively on the style features $X^\perp$ (this is only for notational convenience) and let the style features $X^\perp$ act in a linear way on the image $X$ via a matrix $W \in \mathbb{R}^{p \times q}$ (this is an important assumption without which results are more involved). The core or 'conditionally invariant' features are $X^{ci} \in \mathbb{R}^r$, where in general $r \leq p$ but this is not important for the following. For independent $\varepsilon_Y, \varepsilon_{\mathrm{ID}}, \varepsilon_{X^\perp}, \varepsilon_X$ in $\mathbb{R}, \mathbb{R}^q, \mathbb{R}^r, \mathbb{R}^p$ respectively with positive density on their support and continuously differentiable functions $k_y, k_{\mathrm{ID}}, k_{X^\perp}, k_{X^{ci}}, k_x$,

$$\text{class } Y \leftarrow k_y(D, \varepsilon_Y)$$
$$\text{identifier ID} \leftarrow k_{\mathrm{ID}}(Y, \varepsilon_{\mathrm{ID}})$$
$$\text{core or conditionally invariant features } X^{ci} \leftarrow k_{X^{ci}}(Y, \mathrm{ID})$$
$$\text{style or orthogonal features } X^\perp \leftarrow k_{X^\perp}(D, Y, \varepsilon_{X^\perp}) + \Delta$$
$$\text{image } X \leftarrow k_X(X^{ci}, \varepsilon_X) + WX^\perp.$$

Of these, $Y$, $X$ and ID are observed whereas $D, X^{ci}, \Delta, X^\perp$ and the noise variables are latent.

We assume a logistic regression as a prediction of $Y$ from the image data $X$:

$$f_\theta(x) := \frac{\exp(x^t\theta)}{1 + \exp(x^t\theta)}.$$

Given training data with $m$ samples, we estimate $\theta$ with $\hat\theta$ and use here a logistic loss $\ell_\theta(y_i, x_i) = \log(1 + \exp(-y_i(x_i^t\theta)))$ for training and testing. Some interesting expected losses on test data include

$$L(\theta) = E\Big[\ell\big(Y, f_\theta(X)\big)\big)\Big]$$
$$L_{adv}(\theta) = E\Big[\max_{\Delta \in \mathbb{R}^q} \ell\big(Y, f_\theta(X(\Delta))\big)\Big],$$

where the $X$ in the first loss is a shorthand notation for $X(\Delta = 0)$, that is the images in absence of interventions on the style variables. The first loss is thus a standard logistic loss in absence of adversarial interventions. The second loss is the loss under adversarial style or domain interventions as we allow arbitrarily large interventions on $X^\perp$ here. The corresponding benchmarks are

$$L^* = \min_\theta L(\theta), \text{ and } L^*_{adv} = \min_\theta L_{adv}(\theta).$$

The formulation of Theorem 1 relies on the following assumptions.

**Assumption 1.** *We require the following conditions to hold:*

*(A1) Assume $\Delta$ is sampled from a distribution for training data in $\mathbb{R}^q$ with positive density on an $\epsilon$-ball in $\ell_2$-norm around the origin for some $\epsilon > 0$.*

*(A2) Assume the matrix $W$ has full rank $q$.*

*(A3) Assume $c \geq q$, that is the number $c = m - n$ of counterfactual examples in the samples is at least as large as the dimension of the style variables.*

Regarding (A3): the sampling process is as follows. We collect $n$ independent samples $(y_i, \mathrm{id}_i, \delta_{i,1})$ from a distribution of $(Y, \mathrm{ID}, \Delta)$ that satisfies the constraints above. Then, for $c = m - n$ of the samples we select each time $i \in \{1, \dots, n\}$ at random, keep $(y_i, \mathrm{id}_i)$ fixed (and hence also the realization of $X^\perp$ is fixed) and redraw a new value of $\Delta$ as $\delta_{i,u_i+1}$ if $u_i$ is the current number of counterfactual examples for sample $i$. This leads to $m$ samples in total with in general $n$ distinct values of $(y_i, \mathrm{id}_i)$ and $m_i$ counterfactuals at each sample with corresponding $x_{i,j}$ with $i \in \{1, \dots, n\}$ and $j \in \{1, \dots, m_i\}$.

**Theorem 1.** *Under Assumption 1, with probability 1 with respect to the training data, the pooled estimator has infinite adversarial loss*

$$L_{adv}(\hat{\theta}^{pool}) = \infty.$$

*For the* CORE *estimator, for* $n \to \infty$,

$$L_{adv}(\hat{\theta}^{core}) \to_p L_{adv}^*.$$

An equivalent results can be derived for misclassification loss instead of logistic loss (with infinity replaced by 1).

*Proof.* **First part.** To show the first part, namely that with probability 1,

$$L_{adv}(\hat{\theta}^{pool}) = \infty,$$

we need to show that $W^t \hat{\theta}^{pool} \neq 0$ with probability 1. The reason this is sufficient is as follows: if $W^t \theta \neq 0$, then $L_{adv}(\theta) = \infty$ as we can then find a $v \in \mathbb{R}^q$ such that $\gamma := \theta^t W v \neq 0$. Setting $\Delta_\kappa = \kappa v$ for $\kappa \in \mathbb{R}$, we get $x(\Delta_\kappa)^t \theta = x(\Delta = 0)^t \theta + \kappa \gamma$. Hence $\log(1 + \exp(-x(\Delta_\kappa)^t \theta)) \to \infty$ for either $\kappa \to \infty$ or $\kappa \to -\infty$.

To show that $W^t \hat{\theta}^{pool} \neq 0$ with probability 1, let $\hat{\theta}^*$ be the oracle estimator that is constrained to be orthogonal to the column space of $W$:

$$\hat{\theta}^* = \text{argmin}_{\theta:W^t\theta=0} \, L_n(\theta) \text{ with } L_n(\theta) := \frac{1}{n} \sum_{i=1}^n \ell(y_i, f_\theta(x_i(\Delta_i))). \tag{6}$$

We show $W^t \hat{\theta}^{pool} \neq 0$ by contradiction. Assume hence that $W^t \hat{\theta}^{pool} = 0$. If this is indeed the case, then the constraint $W^t \theta = 0$ in (6) becomes non-active and we have $\hat{\theta}^{pool} = \hat{\theta}^*$. This would imply that taking the directional derivative of the training loss with respect to any $\delta \in \mathbb{R}^p$ in the column space of $W$ should vanish at the solution $\hat{\theta}^*$. Define $r_i(\theta) := (y_i + 1)/2 - f_{\hat{\theta}^*}$. For all $i = 1, \ldots, n$ we have $r_i \neq 0$. The derivative $g(\delta)$ of $L_n(\theta)$ in direction of $\delta$ is proportional to

$$g(\delta) = \frac{1}{n} \sum_{i=1}^n r_i(\hat{\theta}^*) \sum_{j=1}^{m_i} x_{i,j}^t \delta, \tag{7}$$

$x_{i,j} \in \mathbb{R}^p$ is the $j$-th counterfactual for training sample $i$ (with $j \in \{1, \ldots, m_i\}$). Let $x_{i,j}(0) = x_{i,1}(0)$ for $i = 1, \ldots, n$ be the counterfactual training data in absence of any interventions ($\Delta_{i,j} = 0$). Since the interventions only have an effect on the column space of $W$ in $X$, the oracle estimator $\hat{\theta}^*$ is identical under the true training data and the counterfactual training data $x(0)$. Hence, for any $\delta$ in $\mathbb{R}^p$, the derivative $g(\delta)$ in (7) can also be written as

$$g(\delta) = \frac{1}{n} \sum_{i=1}^n r_i(\hat{\theta}^*) \sum_{j=1}^{m_i} x_{i,k}(0)^t \delta. \tag{8}$$

Taking the difference between (7) and (8),

$$\frac{1}{n} \sum_{i=1}^n r_i(\hat{\theta}^*) (\sum_{j=1}^{m_i} (x_{i,j} - x_{i,j}(0))^t \delta) = 0. \tag{9}$$

Now, by the model assumptions, $x_{i,j} - x_{i,j}(0) = W\Delta_{i,j}$. Since $\delta$ is in the column-space of $W$, there exists $u \in \mathbb{R}^q$ such that $\delta = Wu$. then (9) can be written as

$$\frac{1}{n} \sum_{i=1}^n r_i(\hat{\theta}^*) \sum_{j=1}^{m_i} \Delta_{i,j}^t W^t W u = 0 \tag{10}$$

From (A2) we have that the eigenvalues of $W^t W$ are all positive. Also $r_i(\hat{\theta}^*)$ is not a function of the interventions $\Delta_{i,j}$ since, as already argued above, the estimator $\hat{\theta}^*$ is identical whether trained on the original data $x_{i,j}$ or on the counterfactual data $x_{i,j}(0)$. If we condition on $(x_i(0), y_i)$ for $i = 1, \ldots, n$ (that is everything except for the random $\Delta_{i,j}, i = 1, \ldots, n$), then the interventions

$\Delta_{i,j}$ are by (A1) drawn from a continuous distribution. Hence the left hand side of (10) has a continuous distribution, and the probability of the left hand side of (10) being not identically 0 is 1. This completes the proof of the first part by contradiction.

**Second part.** For the second part, we first show that with probability 1, $\hat{\theta}^{core} = \hat{\theta}^*$ with $\hat{\theta}^*$ defined as in (6). Note that the invariant space is for this model the linear subspace $I = \{\theta : W^t\theta = 0\}$. Note that by their respective definitions,

$$\hat{\theta}^* = \operatorname{argmin}_\theta \frac{1}{m} \sum_{i=1}^n \sum_{j=1}^{m_i} \ell(y_i, f_\theta(x_{i,j})) \text{ such that } \theta \in I,$$

$$\hat{\theta}^{core} = \operatorname{argmin}_\theta \frac{1}{m} \sum_{i=1}^n \sum_{j=1}^{m_i} \ell(y_i, f_\theta(x_{i,j})) \text{ such that } \theta \in I_n.$$

By (A2) and (A3), with probability 1, $I_n = \{\theta : W^t\theta = 0\}$ since the number of counterfactuals examples is equal to or exceeds the rank $q$ of $W$ and $X^\perp$ has a linear influence on $X$. Hence with probability 1, we have $I = I_n$ and hence $\hat{\theta}^{core} = \hat{\theta}^*$. We thus need to show that

$$L_{adv}(\hat{\theta}^*) \to_p L_{adv}^*. \tag{11}$$

Since $\hat{\theta}^*$ is in $I$, we have $\ell(y, x(\Delta)) = \ell(y, x(0))$, where $x(0)$ are the previously discussed counterfactual data in the absence of interventions. Hence

$$\hat{\theta}^* = \operatorname{argmin}_\theta \frac{1}{m} \sum_{i=1}^n \sum_{j=1}^{m_i} \ell(y_i, f_\theta(x_{i,j}(0))) \text{ such that } \theta \in I, \tag{12}$$

that is the estimator is unchanged if we use the data without interventions ($\Delta_i = 0$) as training data. Define the population-optimal vector as

$$\theta^* = \operatorname{argmin}_\theta E\big[\max_\Delta \ell(Y, f_\theta(X(\Delta)))\big] \text{ such that } \theta \in I,$$

which can for the same reason be written as

$$\theta^* = \operatorname{argmin}_\theta E\big[\ell(Y, f_\theta(X(\Delta = 0)))\big] \text{ such that } \theta \in I. \tag{13}$$

Hence (12) and (13) can be written as

$$\hat{\theta}^* = \operatorname{argmin}_{\theta:\theta\in I} L_n^{(0)}(\theta) \text{ where } L_n^{(0)}(\theta) := \frac{1}{m} \sum_{i=1}^n \sum_{j=1}^{m_i} \ell(y_i, f_\theta(x_{i,j}(0)))$$

$$\theta^* = \operatorname{argmin}_{\theta:\theta\in I} L^{(0)}(\theta) \text{ where } L^{(0)}(\theta) := E[\ell(Y, f_\theta(X(\Delta = 0)))].$$

Comparing (12) and (13), by uniform convergence of $L_n^{(0)}$ to the population loss $L^{(0)}$ under the assumed sampling where $n$ samples of $(Y, \text{ID})$ are drawn independently then $c = m - n$ samples are redrawn from this empirical sample at random, we have $L^{(0)}(\hat{\theta}^*) \to_p L^{(0)}(\theta^*)$.

By definition of $I$ and $\theta^*$ we have $L_{adv}^* = L_{adv}(\theta^*) = L^{(0)}(\theta^*)$. As $\hat{\theta}^*$ is in $I$, we also have $L_{adv}(\hat{\theta}^*) = L^{(0)}(\hat{\theta}^*)$. Since, from above, $L^{(0)}(\hat{\theta}^*) \to_p L^{(0)}(\theta^*)$, this also implies $L_{adv}(\hat{\theta}^*) \to_p L_{adv}(\theta^*) = L_{adv}^*$. This completes the proof, using the previous fact that $\hat{\theta}^{core} = \hat{\theta}^*$ with probability 1 under (A3). $\square$

# B  ADDITIONAL EXPERIMENTS

## B.1  GENDER CLASSIFICATION

We work with the CelebA dataset (Liu et al., 2015) and consider the problem of classifying whether the person in the image is male or female. We create a confounding by including mostly images of men wearing glasses while the images of women do not include photos of women with glasses. As counterfactuals, we use an image of the same person without glasses if the person is male and

(a) Training examples.

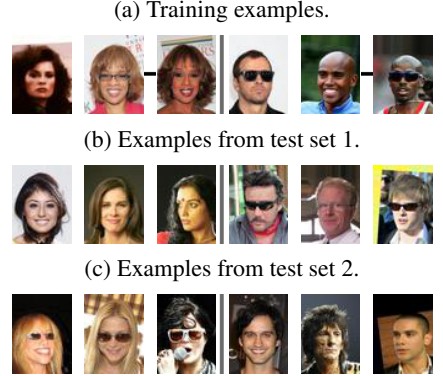

(b) Examples from test set 1.

(c) Examples from test set 2.

Figure B.1: Examples from the CelebA gender dataset.

with glasses if the person is female. We call using an image of the same person as counterfactual "CF setting 2". Examples from the training and test sets are shown in Figure B.2. Test set 1 follows the same distribution as the training set. In test set 2 the association between gender and glasses is flipped: women always wear glasses while men never wear glasses.

In this example, we would like to assess whether the results will differ when (a) training a four-layer CNN (as detailed in Table C.1) end-to-end versus (b) using Inception V3 features and merely retraining the softmax layer. Figure B.2 shows the results for varying numbers of $m$ and $c$—in the left column for training a four-layer CNN; in the right column for using Inception V3 features. Overall, we see the same trends: As $c$ increases, the performance difference between CORE and the pooled estimator becomes smaller. This is due to the fact that $X^\perp$ is binary in this example and, therefore, including counterfactual examples corresponds to data augmentation. Interestingly, the pooled estimator performs worse on test set 2 as $m$ becomes larger. It thus seems to exploit $X^\perp$ to a larger extent as $m$ grows.

### B.2 EYEGLASSES DETECTION: BRIGHTNESS INTERVENTION

As in §5.2 we work with the CelebA dataset and consider the problem of classifying whether the person in the image is wearing eyeglasses. Here we analyze a confounded setting that could arise as follows. Say the hidden common cause of $Y$ and $X^\perp$, $D$ indicates whether the image was taken outdoors or indoors. If it was taken outdoors, then the person wears glasses and the image tends to be brighter. If the image was taken indoors, then the person does not wear glasses and the image tends to be darker. In other words, $X^\perp \equiv brightness$ and the structure of the data generating process is equivalent to the one shown in Figure C.9. Figure B.3a shows examples from the training set. Here, we use as the counterfactual observation the same image (CF setting 1) but with a different brightness. Two alternatives for constructing counterfactual observations in this setting are discussed in §B.2.1. We use $c = 2000$ and $m = 20000$.

For the brightness intervention, we sample the value for the magnitude of the brightness increase resp. decrease from an exponential distribution with mean $\beta = 20$. Specifically, we use ImageMagick[9] to modify the brightness of each image. In the training set and test set 1, we sample the brightness value as $b_{i,j} = 100 + y_i e_{i,j}$ where $e_{i,j} \sim Exp(\beta^{-1})$ and $y_i \in \{-1, 1\}$. $y_i = 1$ corresponds to $y_i \equiv glasses$. We then apply the command `convert -modulate b_ij,100,100 input.jpg output.jpg` to the image. Importantly, since we sample from an exponential distribution, the brightness interventions are quite subtle in many cases as can be seen in Figure B.3a.

Figure B.3c shows misclassification rates for CORE and the pooled estimator on different test sets. Examples from all test sets can be found in Figure B.4. Test set 1 follows the same distribution as the training set. In test set 2 the sign of the brightness intervention is reversed, i.e. images of people with glasses tend to be darker; images of people without glasses tend to be brighter. In test set 3 all images are left unchanged and in test set 4 the brightness of all images is increased. First, we notice that the pooled estimator performs better than CORE on test set 1. This can be explained by

---

[9] https://www.imagemagick.org

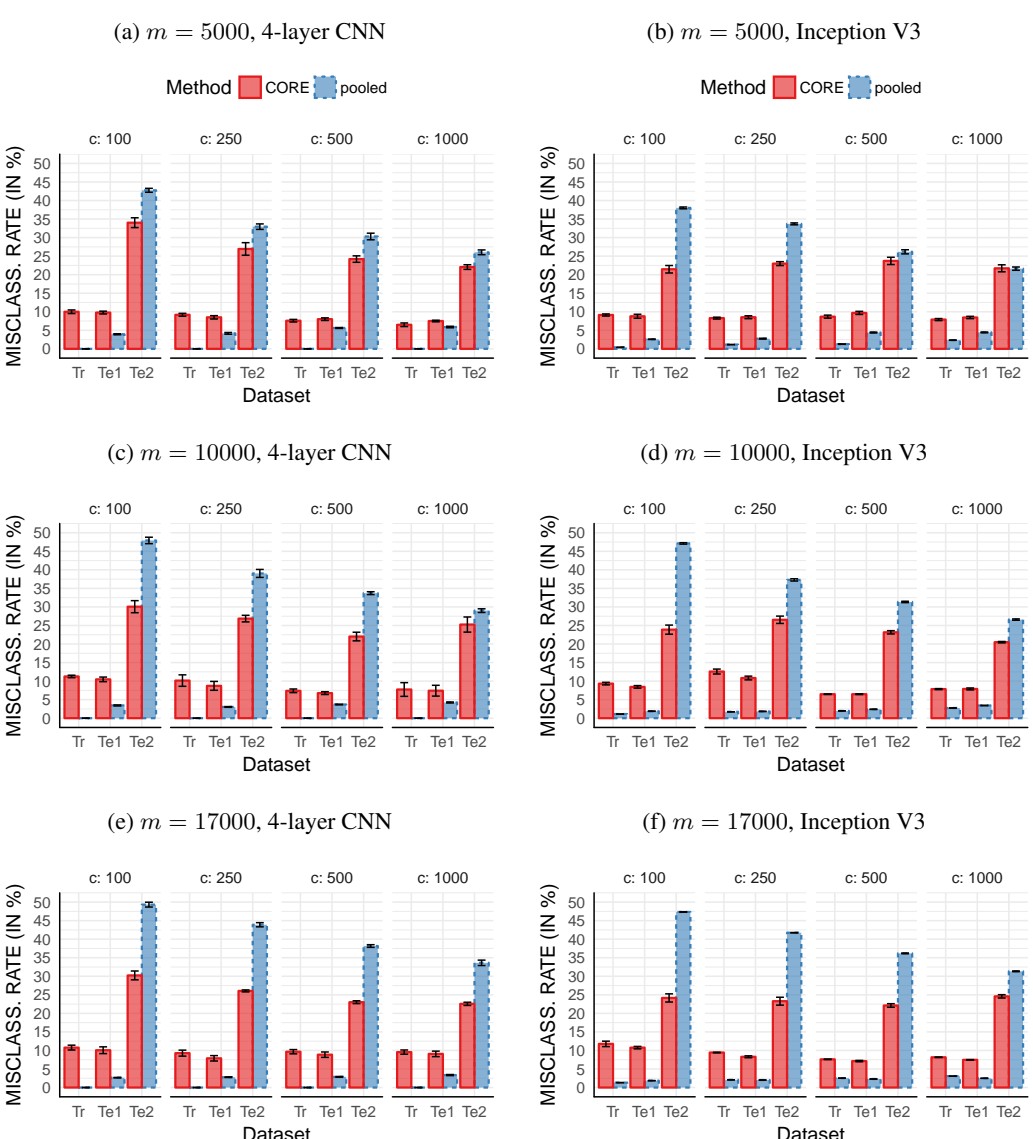

Figure B.2: Misclassification rates for the CelebA gender datasets with varying numbers for $m$ and $c$. The left column shows results for training a four-layer CNN (cf. Table C.1) end-to-end, the right column shows results for using Inception V3 features and retraining the softmax layer.

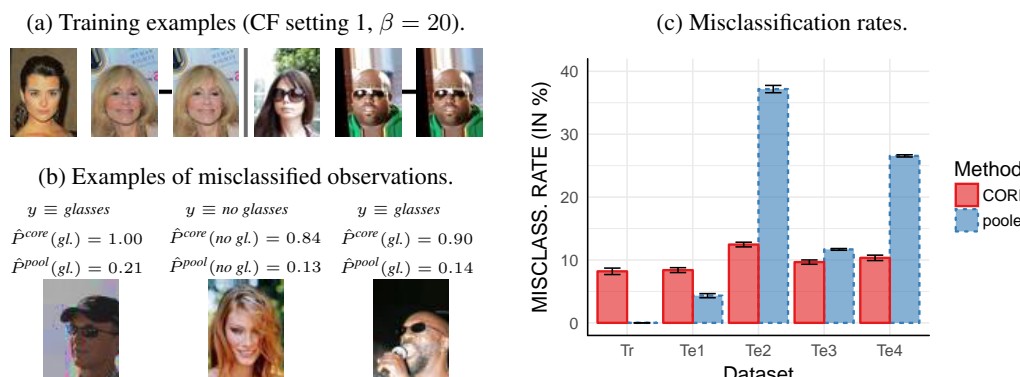

Figure B.3: a) Examples from the CelebA brightness dataset. The first three images from the left have $y \equiv$ *no glasses*; the remaining three images have $y \equiv$ *glasses*. Connected images are counterfactual examples. b) Misclassified examples from the test sets. c) Misclassification rates for $\beta = 20$ and $c = 2000$. Results for different counterfactual settings, $\beta \in \{5, 10, 20\}$ and $c \in \{200, 5000\}$ can be found in Figure B.5.

the fact that it can exploit the predictive information contained in the brightness of an image while CORE is restricted not to do so. Second, we observe that the pooled estimator does not perform well on test sets 2 and 4 as its learned representation seems to use the image's brightness as a predictor for the response which fails when the brightness distribution in the test set differs significantly from the training set. In contrast, the predictive performance of CORE is hardly affected by the changing brightness distributions. Results for $\beta \in \{5, 10, 20\}$ and $c \in \{200, 5000\}$ can be found in Figure B.5.

### B.2.1 COUNTERFACTUAL SETTINGS 2 AND 3

Above we used the same image to create a counterfactual observation by sampling a different value for the brightness intervention. A plausible alternative is to use a different image of the same person as counterfactual. We call this "CF setting 2". For comparison, we also evaluate using an image of a different person as counterfactual as a baseline ("CF setting 3"). Examples from the training sets using CF setting 2 and 3 can be found in Figure B.4.

Results for all counterfactual settings, $\beta \in \{5, 10, 20\}$ and $c \in \{200, 5000\}$ can be found in Figure B.5. We see that using counterfactual setting 1 works best since we could explicitly control that only $X^{\perp} \equiv$ *brightness* varies between counterfactual examples. In counterfactual setting 2, different images of the same person can vary in many factors, making it more challenging to isolate brightness as the factor to be invariant against. Lastly, we see that even grouping images of different persons can still help predictive performance to some degree.

## C EXPERIMENTAL DETAILS AND ADDITIONAL RESULTS FOR EXPERIMENTS INTRODUCED IN §2 AND §5

### C.1 CHOOSING THE TUNING PARAMETER $\lambda$

An open question is how to set the value of the tuning parameter $\tau$ in Eq. (4) or the penalty $\lambda$ in the Lagrangian form. Figure C.6 shows the misclassification rates of CORE on the subsampled and augmented AwA2 dataset as a function of the penalty $\lambda$. We see that performance is not very sensitive to the choice of $\lambda$.

### C.2 GROUPING PHOTOS OF THE SAME PERSON: BETTER PREDICTIVE PERFORMANCE

Here, we show further results for the experiment introduced in §2.1. We vary the number of identities included in the training data set $n \in \{10, 20, 40, 80, 160\}$. This results in total sample sizes $m$ ranging from 321 for $n = 10$ to 4386 for $n = 160$, implying that the average number of counterfactual observations per person varies between 27 and 32. Figure C.7b shows the misclassification

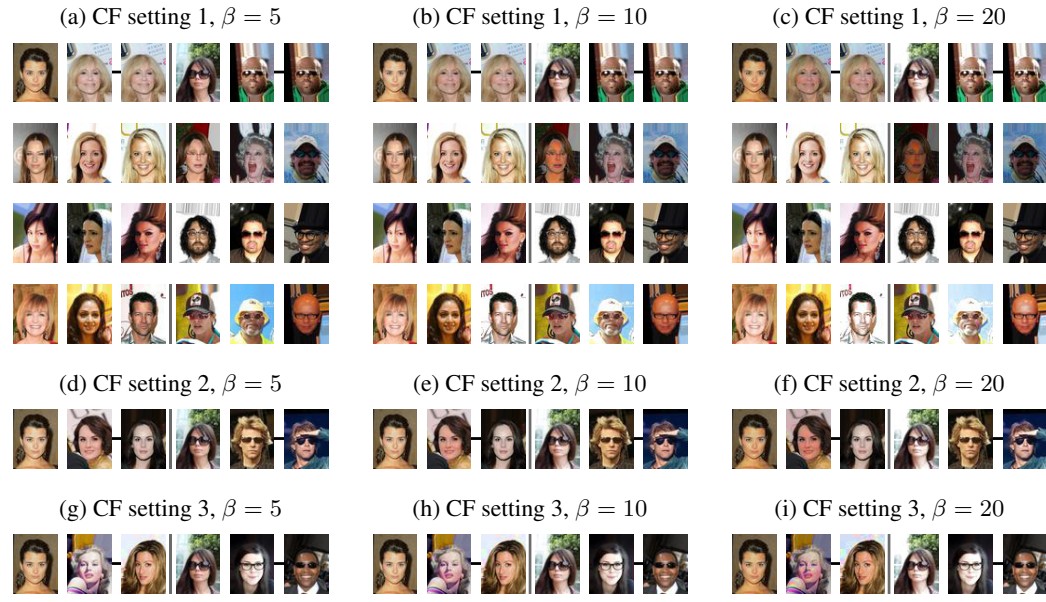

Figure B.4: Examples from the CelebA brightness datasets, counterfactual settings 1–3 with $\beta \in \{5, 10, 20\}$. In all rows, the first three images from the left have $y \equiv$ *no glasses*; the remaining three images have $y \equiv$ *glasses*. Connected images are counterfactual examples. In panels (a)–(c), row 1 shows examples from the training set, rows 2–4 contain examples from test sets 2–4, respectively. Panels (d)–(i) show examples from the respective training sets.

rates for the test set which consists of 5000 examples. We see that CORE helps predictive performance compared to the estimator which just pools all images, notably when $n$ is very small. It thus successfully mitigates the effect of potential confounders arising due to small sample sizes. As $n$ and $m$ increase the performance of CORE and the pooled estimator become comparable—the larger sample sizes ensure that fewer confounding factors are present in the training data and exploited by the pooled estimator.

## C.3 GROUPING AUGMENTED IMAGES BY ORIGINAL: MORE SAMPLE EFFICIENT

Here, we show further results for the experiment introduced in §2.2. We vary the number of augmented training examples $c$ from 100 to 5000 for $n = 10000$ and $c \in \{100, 200, 500, 1000\}$ for $n = 1000$. The degree of the rotations is sampled uniformly at random from $[35, 70]$. Figure C.8 shows the misclassification rates. Test set 1 contains rotated digits only, test set 2 is the usual MNIST test set. We see that the misclassification rates of CORE are always lower on test set 1, showing that it makes data augmentation more efficient. For $n = 1000$, it even turns out to be beneficial for performance on test set 2.

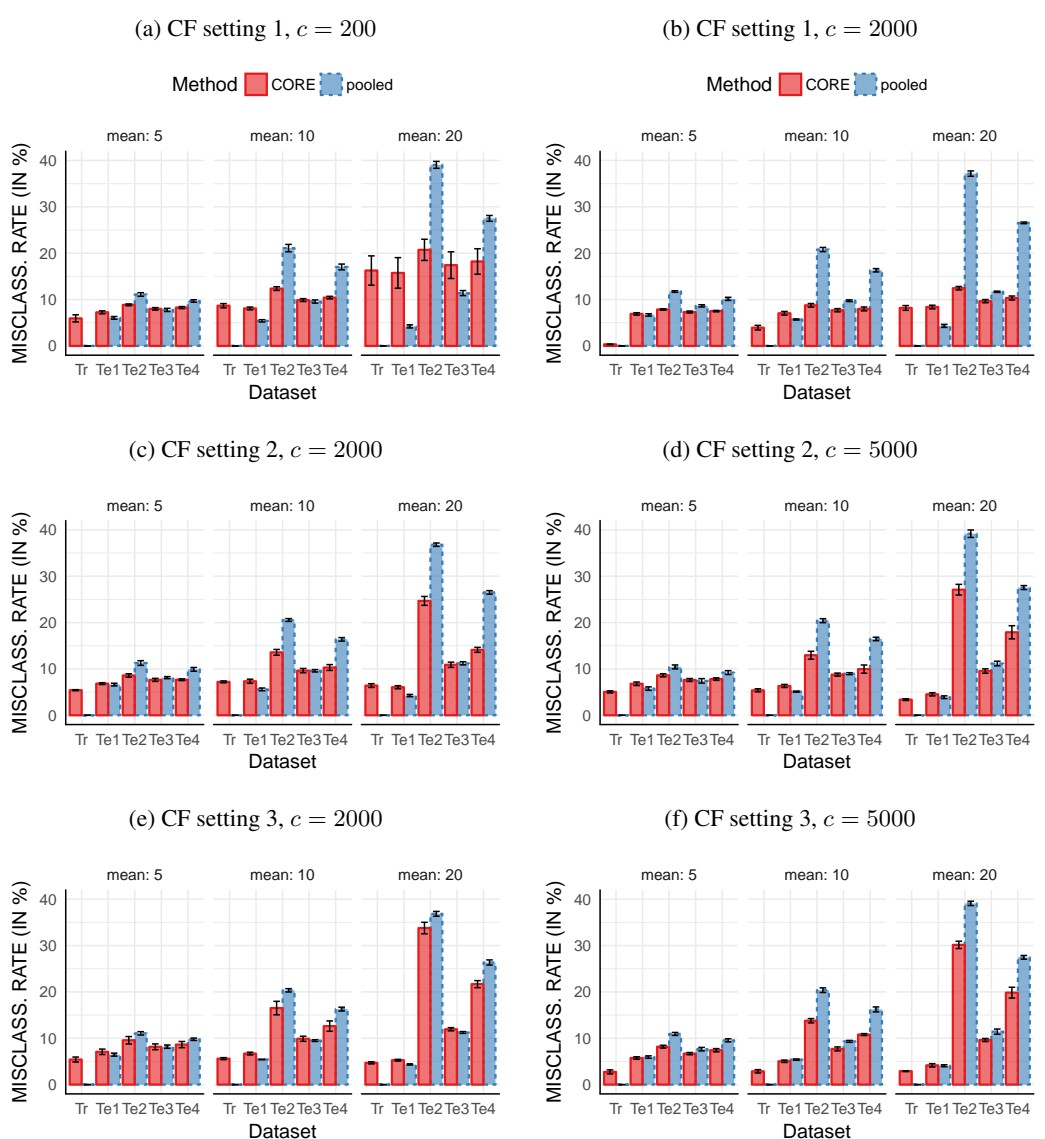

Figure B.5: Misclassification rates for the CelebA brightness datasets, counterfactual settings 1–3 with $c \in \{200, 2000, 5000\}$ and the mean of the exponential distribution $\beta \in \{5, 10, 20\}$.

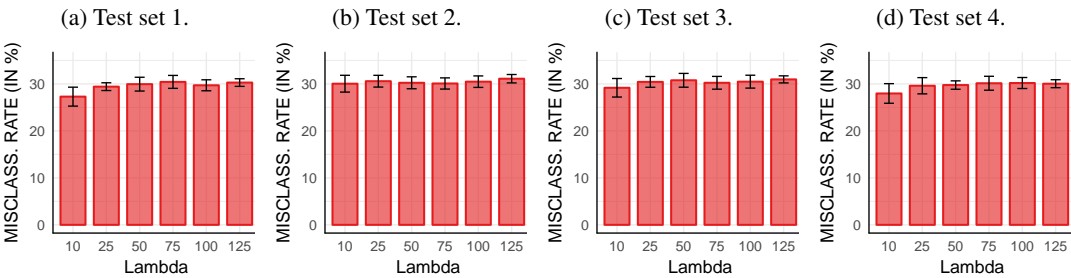

Figure C.6: Misclassification rates of CORE on the subsampled and augmented AwA2 dataset as a function of the penalty $\lambda$. The outcome does not depend strongly on the chosen value.

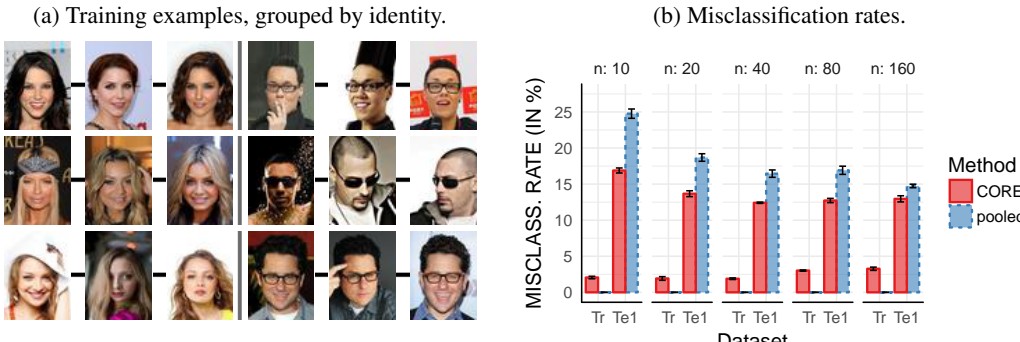

(a) Training examples, grouped by identity.   (b) Misclassification rates.

Figure C.7: a) Examples from the subsampled CelebA dataset. In each row, the first three images from the left have $y \equiv$ *no glasses*; the remaining three images have $y \equiv$ *glasses*. Connected images are counterfactual examples. b) Misclassification rates for different numbers of identities, included in the training data.

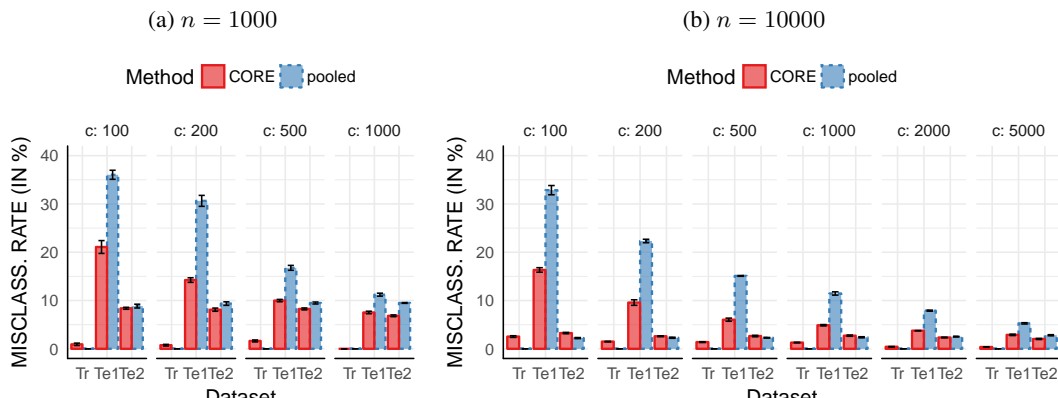

(a) $n = 1000$   (b) $n = 10000$

Figure C.8: Data augmentation setting: Misclassification rates for MNIST and $S \equiv$ *rotation*. In test set 1 all digits are rotated by a degree randomly sampled from $[35, 70]$. Test set 2 is the usual MNIST test set.

(a) Examples from test sets 1–3.       (b) Misclassification rates.

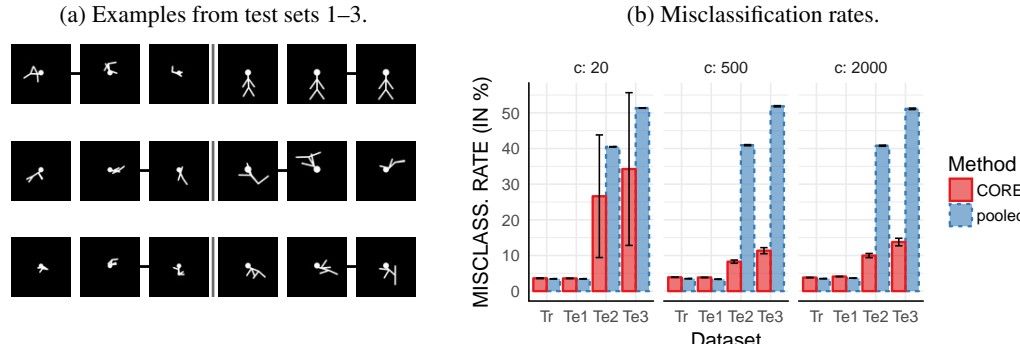

Figure C.10: a) Examples from the stickmen test set 1 (row 1), test set 2 (row 2) and test sets 3 (row 3). In each row, the first three images from the left have $y \equiv$ *child*; the remaining three images have $y \equiv$ *adult*. Connected images are counterfactual examples. b) Misclassification rates for different numbers of counterfactual examples.

## C.4 STICKMEN IMAGE-BASED AGE CLASSIFICATION

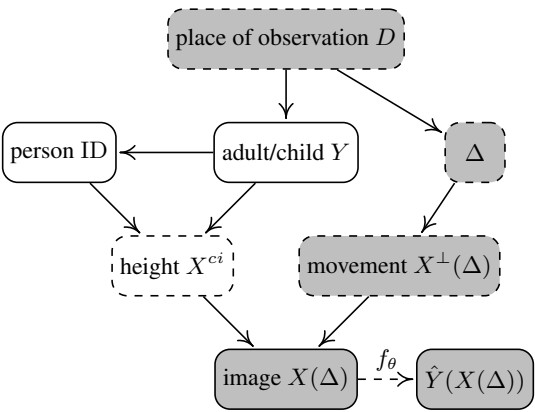

Figure C.9: Data generating process for the stickmen example.

Here, we show further results for the experiment introduced in §5.1. Figure C.10b shows results for different numbers of counterfactual examples. For $c = 20$ the misclassification rate of CORE estimator has a large variance. For $c \in \{50, 500, 2000\}$, the CORE estimator shows similar results. Its performance is thus not sensitive to the number of counterfactual examples, once there are sufficiently many counterfactual observations in the training set. The pooled estimator fails to achieve good predictive performance on test sets 2 and 3 as it seems to use "movement" as a predictor for "age".

## C.5 EYEGLASSES DETECTION: IMAGE QUALITY INTERVENTION

Here, we show further results for the experiment introduced in §5.2. Specifically, we consider interventions of different strengths by varying the mean of the quality intervention in $\mu \in \{30, 40, 50\}$. As in §B.2, we use ImageMagick, this time to modify the image quality. In the training set and in test set 1, we sample the image quality value as $q_{i,j} \sim \mathcal{N}(\mu, \sigma = 10)$ and apply the command `convert -quality q_ij input.jpg output.jpg` if $y_i \equiv$ *glasses*. If $y_i \equiv$ *no glasses*, the image is not modified. In test set 2, the above command is applied if $y_i \equiv$ *no glasses* while images with $y_i \equiv$ *glasses* are not changed. In test set 3 all images are left unchanged and in test set 4 the command is applied to all images, i.e. the quality of all images is reduced.

We run experiments for counterfactual settings 1–3 and for $c = 5000$. Figure C.11 shows examples from the respective training and test sets and Figure C.12 shows the corresponding misclassification rates. Again, we observe that counterfactual setting 1 works best while there are only small differences in predictive performance between counterfactual settings 2 and 3. Interestingly, there is a

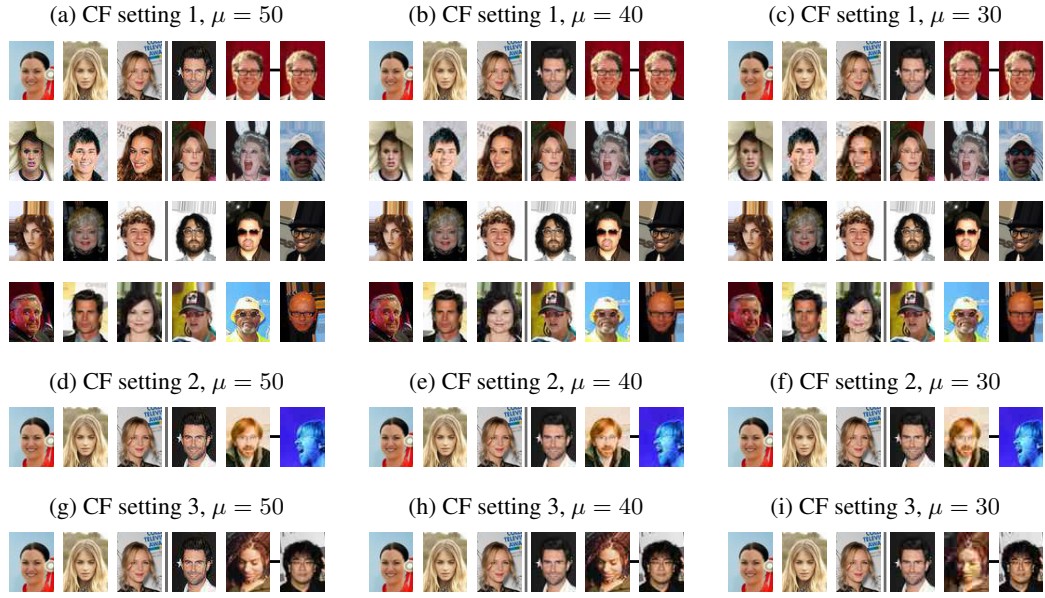

(a) CF setting 1, $\mu = 50$     (b) CF setting 1, $\mu = 40$     (c) CF setting 1, $\mu = 30$

(d) CF setting 2, $\mu = 50$     (e) CF setting 2, $\mu = 40$     (f) CF setting 2, $\mu = 30$

(g) CF setting 3, $\mu = 50$     (h) CF setting 3, $\mu = 40$     (i) CF setting 3, $\mu = 30$

Figure C.11: Examples from the CelebA image quality datasets, counterfactual settings 1–3 with $\mu \in \{30, 40, 50\}$. In all rows, the first three images from the left have $y \equiv no\ glasses$; the remaining three images have $y \equiv glasses$. Connected images are counterfactual examples. In panels (a)–(c), row 1 shows examples from the training set, rows 2–4 contain examples from test sets 2–4, respectively. Panels (d)–(i) show examples from the respective training sets.

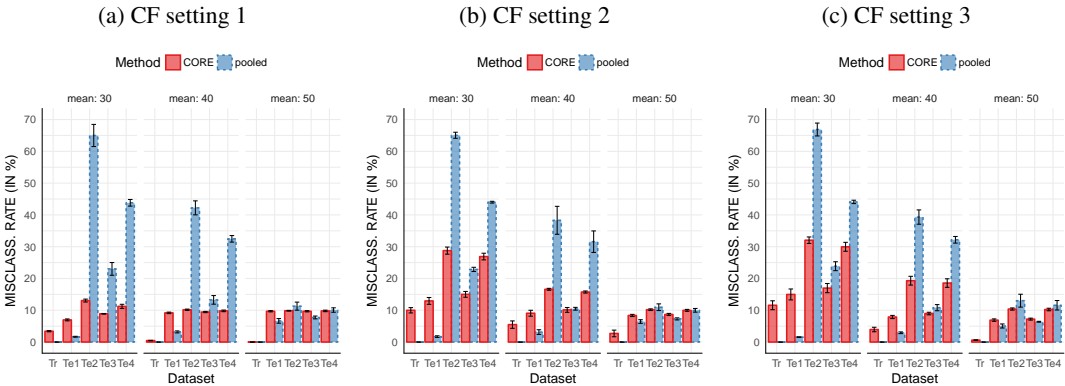

(a) CF setting 1     (b) CF setting 2     (c) CF setting 3

Figure C.12: Misclassification rates for the CelebA image quality datasets, counterfactual settings 1–3 with $c = 5000$ and the mean of the Gaussian distribution $\mu \in \{30, 40, 50\}$.

large performance difference between $\mu = 40$ and $\mu = 50$ for the pooled estimator. Possibly, with $\mu = 50$ the image quality is not sufficiently predictive for the target.

## C.6 ELMER THE ELEPHANT

The color interventions for the experiment introduced in §5.3 are created as follows. In the training set, if $y_i \equiv elephant$ we apply the following ImageMagick command only for the counterfactual examples `convert -modulate 100,0,100 input.jpg output.jpg`, producing a grayscale image. In test set 1, all images are left unchanged. In test set 2, the above command is applied if $y_i \equiv horse$; if $y_i \equiv elephant$ we sample $c_{i,j} \sim \mathcal{N}(\mu = 20, \sigma = 1)$ and apply `convert -modulate 100,100,100-c_ij input.jpg output.jpg` to the image. In test set 4, the

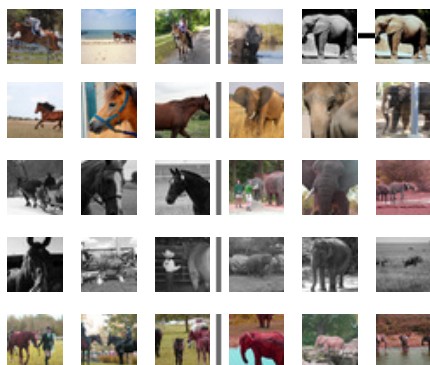

Figure C.13: Examples from the subsampled and augmented AwA2 dataset. Row 1 shows examples from the training set, rows 2–5 show examples from test sets 1–4, respectively.

latter command is applied to all images. It rotates the colors of the image, in a cyclic manner[10]. In test set 3, all images are changed to grayscale.

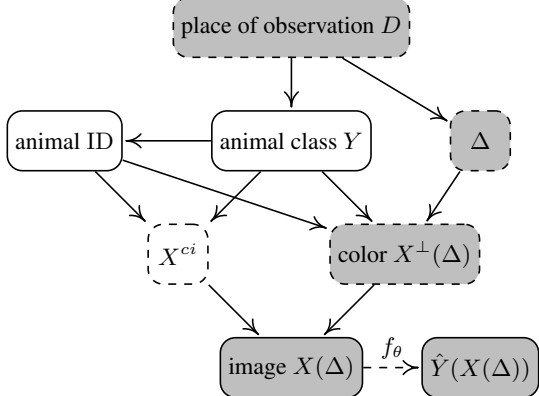

Figure C.14: Data generating process for the Elmer the elephant example.

## C.7 NETWORK ARCHITECTURES

We implemented the considered models in TensorFlow (Abadi et al., 2015). The model architectures used are detailed in Table C.1. CORE and the pooled estimator thus use the same network architecture and training procedure; merely the loss function differs by the counterfactual regularization term. In all experiments we use the Adam optimizer (Kingma & Ba, 2015).

All experimental results are based on training the respective model five times (using the same data) to assess the variance due to the randomness in the training procedure.

In each epoch of the training, the training data $x_{i,\cdot}, i = 1, \ldots, n$ is randomly shuffled, keeping the counterfactual observations $x_{i,j}, j = 1, \ldots, m_i$ together to ensure that mini batches will contain counterfactual observations. In all experiments the mini batch size is set to 120. For small $c$ this implies that not all mini batches contain counterfactual observations, making the optimization more challenging.

---

[10]For more details, see `http://www.imagemagick.org/Usage/color_mods/#color_mods`.

| Dataset | Optimizer | Architecture | |
|---|---|---|---|
| MNIST | Adam | Input | $28 \times 28 \times 1$ |
| | | CNN | Conv $5 \times 5 \times 16, 5 \times 5 \times 32$ |
| | | | (same padding, strides $= 2$, ReLu activation), |
| | | | fully connected, softmax layer |
| Stickmen | Adam | Input | $64 \times 64 \times 1$ |
| | | CNN | Conv $5 \times 5 \times 16, 5 \times 5 \times 32, 5 \times 5 \times 64, 5 \times 5 \times 128$ |
| | | | (same padding, strides $= 2$, leaky ReLu activation), |
| | | | fully connected, softmax layer |
| CelebA (all experiments using CelebA) | Adam | Input | $64 \times 48 \times 3$ |
| | | CNN | Conv $5 \times 5 \times 16, 5 \times 5 \times 32, 5 \times 5 \times 64, 5 \times 5 \times 128$ |
| | | | (same padding, strides $= 2$, leaky ReLu activation), |
| | | | fully connected, softmax layer |
| AwA2 | Adam | Input | $32 \times 32 \times 3$ |
| | | CNN | Conv $5 \times 5 \times 16, 5 \times 5 \times 32, 5 \times 5 \times 64, 5 \times 5 \times 128$ |
| | | | (same padding, strides $= 2$, leaky ReLu activation), |
| | | | fully connected, softmax layer |

Table C.1: Details of the model architectures used.

