# OpenReview forum: "Grouping-By-ID: Guarding Against Adversarial Domain Shifts"
_ICLR.cc/2018/Conference — Reject_

### Official Review · AnonReviewer1 · 2017-11-27
**Lacks sufficient comparisons to other similar regularizers**

**Rating:** 4
**Confidence:** 5

**Review:**

Proposal is to restrict the feasible parameters to ones that have produce a function with small variance over pre-defined groups of images that should be classified the same. As authors note, this constraint can be converted into a KKT style penalty with KKT multiplier lambda.  Thus this is very  similar to other regularizers that increase smoothness of the function, such as total variation or a graph Laplacian defined with graph edges connecting the examples in each group, as well as manifold regularization (see e.g. Belkin, Niyogi et al. JMLR).  Heck, in practie ridge regularization will also do something similar for many function classes.

Experiments didn't compare to any similar smoothness regularization (and my preferred would have been a comparison to graph Laplacian or total variation on graphs formed by the same clustered examples). It's also not clear either how important it is that they hand-define the groups over which to minimize variance or if just generally adding smoothness regularization would have achieved the same results.   That made it hard to get excited about the results in a vacuum.

Would this proposed strategy have thwarted the Russian tank legend problem? Would it have fixed the Google gorilla problem? Why or why not?

Overall, I found the writing a bit bombastic for a strategy that seems to require the user to hand-define groups/clusters of examples.

Page 2: calling additional instances of the same person “counterfactual observations” didn’t seem consistent with the usual definition of that term… maybe I am just missing the semantic link here, but this isn't how we usually use the term counterfactual in my corner of the field.

Re: “one creates additional samples by modifying…” be nice to quote more of the early work doing this, I believe the first work of this sort was Scholkopf’s, he called it “virtual examples” and I’m pretty sure he specifically did it for rotation MNIST images (and if not exactly that, it was implied).  I think the right citation is “Incorporating invariances in support vector learning machines
“ Scholkopf, Burges, Vapnik 1996, but also see Decoste * Scholkopf 2002 “Training invariant support vector machines.”

---

> ### Author Response · Authors · 2017-12-06
> **Clarifications regarding the relation to different smoothness regularization and terminology**
>
> Thank you for your helpful comments and sharing your concerns.
> We think your concerns are partially based on sloppy writing on our part. We changed the version of the manuscript accordingly.
>
> 1. Comparison to ridge regularization
>
> It was hidden in the supplementary material, but the baseline of the pooled estimator is already computed with a ridge penalty. We have now made this more explicit in Section 4.4.1.
>
> 2. Comparison to different smoothness regularization (preferred one a graph Laplacian or total variation on graphs formed by the same clustered examples)
>
> Again, this is due to our writing for which we apologize. In fact, the proposed estimator is exactly equal to the regularization you propose, namely a smoothness regularization if using the graph Laplacian. We have clarified this in Section 4.5 and also mention it in the abstract now.
>
> The underlying graph is as you propose: all examples that share the same identifier ID are fully connected so that there are n connectivity components in the graph. All edges have a unit weight. Using this graph Laplacian is then equivalent to penalizing the variance across all connected examples that we mentioned in the first version of the text. In most examples we only observe a few pairs of samples that share the same identifier and the graph is then the empty graph with the exception of a few isolated edges.
>
> Our proposed estimator is indeed very simple. The simplicity could be argued to be a strength or weakness. Our point to make in this paper is that (a) we motivate why this specific form of penalty makes sense in a causal context, (b) we derive some theoretical results for logistic regression where we can show the right properties of the estimator for adversarial domain changes and (c) show that the approach works very well empirically when trying to protect against domain changes in a variety of settings.
>
> Furthermore, in the supplement “B.2.1 Counterfactual settings 2 and 3” we compare grouping by the same object vs. grouping different objects and show that grouping by the same object is empirically much more desirable (as expected). Specifically, in Figures B.4 and B.5 we compare if we use as groupings:
> (1) the same picture under different brightness settings (this is a perhaps making the problem a bit too simple)
> (2) pictures of the same person (with varying brightness but also varying backgrounds, postures etc.)
> (3) as comparison: pictures of different people
> Setting 2 is what we have in mind as a realistic application scenario and it can be seen that it performs better than using Setting 3, where we group pictures of different people. Setting 1 produces the best results but then we have in this setting more or less pre-specified brightness as the style variable we want to achieve invariance against.
>
> 3. Russian tank legend/ Google gorilla problem.
>
> We explicitly mimic the Russian tank legend problem (image quality) in Section 5.2 and show that counterfactual regularization performs much better than the standard approach. It is interesting to note for this example that it is enough to have two images of a Russian tank (if translating setting of 5.2 back to the Russian tank problem) in bad and very bad quality. There does not need to be a picture of Russian tank in good quality in the database (although it would not hurt). Having both these examples of the same tank in bad and even worse quality connected as ‘counterfactuals’ (if we can still use the term) leads to automatic exclusion of image quality as a feature for classification.
>
> The Google gorilla problem is more involved. However, we do show an example where the style features correspond to color in Section 5.3.
>
> 4. Hand-define groups/clusters of examples.
>
> To bring the paper more in line with notation in Gong et al, we have now explicitly included the identifier ID that is used for grouping in the causal graph. The identifier ID relates to the person in an image or a specific animal, a specific house or different objects. The core estimator needs information about such an identifier variable ID. If the information is present in a dataset, we would not call it hand-picked even though we acknowledge that not all datasets contain such information. Assuming that we have an identifier ID saves us, on the other hand, from having to know the domain D explicitly as in Gong et al. so the approaches are complementary in this sense.
>
> 5.  Terminology
>
> Thanks. The same concern was raised by the second reviewer. We have tried to explain the reasoning in the new version (and in the answer above to the second reviewer). Having said this, we would be happy to delete the term counterfactual if it is seen as too “bombastic” or too confusing.
>
> 6. Virtual examples
>
> Thank you for the references which are now included.

---

### Official Review · AnonReviewer3 · 2017-11-27
**Potentially interesting, but fails to mention very related work**

**Rating:** 7
**Confidence:** 3

**Review:**

The paper discusses ways to guard against adversarial domain shifts with so-called counterfactual regularization. The main idea is that in several datasets there are many instances of images for the same object/person, and that taking this into account by learning a classifier that is invariant to the superficial changes (or “style” features, e.g. hair color, lighting, rotation etc.) can improve the robustness and prediction accuracy. The authors show the benefit of this approach, as opposed to the naive way of just using all images without any grouping, in several toy experimental settings.

Although I really wanted to like the paper, I have several concerns. First and most importantly, the paper is not citing several important related work. Especially, I have the impression that the paper is focusing on a very similar setting (causally) to the one considered in  [Gong et al. 2016] (http://proceedings.mlr.press/v48/gong16.html), as can be seen from Fig. 1. Although not focusing on classification directly, this paper also tries to a function T(X) such that P(Y|T(X)) is invariant to domain change. Moreover, in that paper, the authors assume that even the distribution of the class can be changed in the different domains (or interventions in this paper).
Besides, there are also other less related papers, e.g. http://proceedings.mlr.press/v28/zhang13d.pdf, https://www.aaai.org/ocs/index.php/AAAI/AAAI15/paper/view/10052/0, https://arxiv.org/abs/1707.09724, (or potentially https://arxiv.org/abs/1507.05333 and https://arxiv.org/abs/1707.06422), that I think may be mentioned for a more complete picture. Since there is some related work, it may be also worth to compare with it, or use the same datasets.

I’m also not very happy with the term “counterfactual”. As the authors mention in footnote, this is not the correct use of the term, since counterfactual means “against the fact”. For example, a counterfactual query is “we gave the patient a drug and the patient died, what would have happened if we didn’t give the drug?” In this case, these are just different interventions on possibly the same object. I’m not sure that in the practical applications one can assure that the noise variables stay the same, which, as the authors correctly mention, would make it a bit closer to counterfactuals. It may sound pedantic, but I don’t understand why use the wrong and confusing terminology for no specific reason, also because in practice the paper reduces to the simple idea of finding a classifier that doesn’t vary too much in the different images of the single object.

**EDIT**: I was satisfied with the clarifications from the authors and I appreciated the changes that they did with respect to the related work and terminology, so I changed my evaluation from a 5 (marginally below threshold) to a 7 (good paper, accept).

---

> ### Author Response · Authors · 2017-12-06
> **Clarifications regarding related work, problem settings and terminology**
>
> Thanks for the helpful comments.
>
> 1. Relationship to “Domain Adaptation with Conditional Transferable Components” by Gong et al.
>
> Thanks for the reference which was also highlighted by the first reviewer. It is inexcusable that we omitted the reference. However, please see answer to point 2 of the first reviewer.  Domain D is latent in our approach while it is observed in Gong et al. In contrast we have to observe the identifier ID that is used for the grouping of samples. The approaches thus share the same goal but work on different datasets (we could not run the core estimator on their datasets as the identifier ID is missing and --vice versa-- the approach of Gong et al would not work on our examples as there is no explicit domain variable in the data).
>
> 2. Terminology
>
> Thanks for the comment and concern regarding the use of the term “counterfactual”. We acknowledge that the term is used in a perhaps non-standard way. We certainly did not want to confuse and have rewritten Section 4.4 accordingly.
>
> In a standard medical example, let Z be health outcome and T the treatment under confounders U (either observed or not). A counterfactual is then for example Z(T=0), the health outcome if no treatment is taken, if in truth treatment has been taken (T=1) and we observed Z(T=1) in the Neyman-Rubin potential outcome notation. We think you criticised that our notation deviates from this standard setting. Let us explain why we used the notation.
>
> We highlight now in Section 4.4 that in the model of Figure 2, the core or “conditionally invariant” features (if using the terminology of Gong) are functions of the class Y and identifier ID only, while the image X=X(Y,ID,Delta)  is a function of class Y, identifier ID,  and the style interventions Delta.
> If we fix (Y,ID)=(y,id), we can observe an image X(y,id,Delta) under different style interventions Delta. In this sense, the images X(y,id,Delta_1), X(y,id,Delta_2),... observed for a fixed (Y,ID)=(y,id) form counterfactuals, just as Z(T=0) and Z(T=1) would form counterfactuals in the medical setting if we fix all confounders for (T,Z). The treatment interventions T are thus set equal to the style interventions Delta in the image setting here.  The image setting is clearly different to the medical example in two ways.
> (i) We would like style interventions Delta in general to have no appreciable effect on the predicted label while in a medical setting we would like treatment T to have an effect as large as possible. (ii) The images X(y,id,Delta) under different style interventions Delta are observable but the different health outcomes Z(T) for different treatments T are in general not all observable as we cannot fix the confounders in practice. We thought this difference is perhaps interesting to note in this context, even if it might appear trivial. We certainly did not want to sow confusion or use the term with no specific reason. We deleted the term “counterfactual” from the title already and would be happy to use different terminology altogether if you still feel that the term is too misleading in this context.

---

### Official Review · AnonReviewer2 · 2017-11-28
**The proposed methods seems useful but novelty seems limited**

**Rating:** 5
**Confidence:** 4

**Review:**

This paper aims at robust image classification against adversarial domain shifts. In the used model, there are two types of latent features, "core" features and "style" features, and the goal is to achieved by avoiding using the changing style features. The proposed method, which makes use of grouping information, seems reasonable and useful.

It is nice that the authors use "counterfactual regularization". But I failed to see a clear, new contribution of using this causal regularization, compared to some of the previous methods to achieve invariance (e.g., relative to translation or rotation). For examples of such methods, one may see the paper "Transform Invariant Auto-encoder" (by Matsuo et al.) and references therein.

The data-generating process for the considered model, given in Figure 2, seems to be consistent with Figure 1 of the paper "Domain Adaptation with Conditional Transferable Components" (by Gong et al.). Perhaps the authors can draw the connection between their work and Gong et al.'s work and the related work discussed in that paper.

Below are some more detailed comments. In Introduction, it would be nice if the authors made it clear that "Their high predictive accuracy might suggest that the extracted latent features and learned representations resemble the characteristics our human cognition uses for the task at hand." Why do the features human cognition uses give an optimal predictive accuracy? On page 2, the authors claimed that "These are arguably one reason why deep learning requires large sample sizes as large sample size is clearly not per se a guarantee that the confounding effect will become weaker." Could the authors give more detail on this? A reference would be appreciated.

---

> ### Author Response · Authors · 2017-12-06
> **Clarifications regarding related work and problem settings**
>
> Thank you for the helpful feedback. We would like to address and clarify the following points.
>
> 1. Relationship to “Transform Invariant Auto-encoder” by Matsuo et al.
>
> We have included the reference in the new version. Besides their work being on autoencoders (which is clearly related to our classification setting), one crucial difference is in our view that the style variable is pre-defined in Matsuo et al. and they only show the instance of shift invariance. In our manuscript, the style variable could be background of an image, color, image quality or any combination of these and, most importantly, the style variable is not pre-defined. Instead we use the grouping variable (called ID in the new version for more succinct notation--see Figure 2) to exclude these style features.
> Even if the notion of transformation would be made much wider in Matsuo et al. in the cost term in Section III.A, another crucial difference is that our work is able to achieve invariance in the presence of confounding, which is not discussed at all in Matsuo et al. The situation arises if the distribution of the style features differs conditional on the class label. As such, the situation is more naturally dealt with in a classification framework as the class label Y does not even appear typically in an auto-encoder setting.
>
> A bit more detail: in the autoencoder setting considered by Matsuo et al., the transform invariant autoencoder generates a “typical spatial subpattern”. In the presence of confounding, there is no guarantee that the transform invariant autoencoder will be able to achieve the desired invariance as the “typical spatial subpattern” will be subject to the same bias as the input data, arising through the confounding. Concretely, consider applying the transform invariant autoencoder to the stickmen dataset from Section 5.1 in our work. We have essentially two groups of images in the training data: moving children and non-moving adults (with very few non-moving children and moving adults). In the absence of a classification setting and class label Y we cannot resolve the confounding: should we sum over all children and then over all adults in (2) in Matsuo et al (hence ignoring movement)  or sum over not moving and then all non-moving pictures (hence ignoring age)?  Even if we decide (more or less arbitrarily in absence of a class label) on the former: while the transform variance term would ensure that, say, two input images of children map to the same output image, the restoration error term would reproduce the bias stemming from the confounding. That is, images of children would still be associated with large movements in the reconstruction while adults would still be associated with small or no movement. If one would then try to classify “adult vs. child” from the pooled reconstructed images, the performance is going to be very similar compared to using the original images---that is, the learned estimator would include `movement’ in its representation and therefore, it would not be robust to adversarial domain shifts, arising through interventions on the style feature `movement’.
>
> Finally, we would also like to highlight that we contribute a theoretical analysis to show robustness to adversarial domain shifts.
>
> 2. Relationship to “Domain Adaptation with Conditional Transferable Components” by Gong et al.
>
> Thanks for the reference which was also highlighted by the second referee. It is inexcusable that we left this reference out in the first version. The settings in both papers are very similar and we have used the revision to make the notation as similar as possible (see for example Figure 2 in the new version). This also led us to include the grouping or identifier variable ID explicitly to make the differences to related approaches more succinct. We discuss the relationship now extensively in the new version. While the goal is similar or even identical,  the main difference is that we use a different data basis for the estimators:
> In Gong et al. the domain variable D can be observed but no equivalent of our identifier ID is available (ID is latent).
> Here we assume domain D is latent but we can observe the identifier ID (at least for a small fraction of samples), where the identifier ID can for example be the person in an image (while the class Y might be whether the person is wearing glasses or not).
> As a result, our methodologies are quite different even if they share the same goal. The differences are discussed in Section 3 and 4.2 in the new version.

---

> ### Author Response · Authors · 2017-12-06
> **Answer to detailed comments**
>
> The first sentence was not meant to claim that human cognition yields optimal predictive accuracy but we can see the misunderstanding it can cause. We deleted it in the new version.
> Lastly, more details regarding the following sentence were requested: “These are arguably one reason why deep learning requires large sample sizes as large sample sizes tend to ensure that the effect of the confounding factors averages out (although a large sample size is clearly not per se a guarantee that the confounding effect will become weaker).” If one only considers a small sample of images, there will be many confounding factors that might be picked up by the classifier. For instance, consider the example of classifying images of dogs versus cats. Now, suppose a dog image always shows a dog on a green lawn while cats are always shown indoors. Thus, the green lawn would be picked up as being predictive for the class “dog”. As the sample size increases and the images come from perhaps more diverse set of sources, more dog and cat images are collected in more diverse environments such that the confounding effect averages out. However, in some cases the data collection might introduce a systematic, persistent confounding: In the Russian tank example, even millions of images of Russian resp. American tanks would not have helped, provided that the quality of American tank images would have always been much better than that of Russian images. As an example, in ImageNet almost all instances of the class “rugby ball” are images showing a rugby ball together with a rugby field. Even though ImageNet is a large dataset, confounding effects can still be present.

---

### Decision · Program_Chairs · 2018-01-29
**ICLR 2018 Conference Acceptance Decision**

**Decision:**

Reject

**Comment:**

The paper proposes a method to robustify neural networks which is an important problem. They uses ideas from causality and create a model that would only depend on "stable" features ignoring the easy to manipulate ones. The paper has some interesting ideas, however, the main concern is regarding insufficient comparison to existing literature. One of the reviewers also has concerns regarding novelty of the approach.